# SANA 1.5: Efficient Scaling of Training-Time and Inference-Time Compute in Linear Diffusion Transformer

**Enze Xie**[* 1]  **Junsong Chen**[* 1]  **Yuyang Zhao**[† 1]  **Jincheng Yu**[† 1]  **Ligeng Zhu**[† 1]  **Yujun Lin**[2]  **Zhekai Zhang**[2]
**Muyang Li**[2]  **Junyu Chen**[3]  **Han Cai**[1]  **Bingchen Liu**[4]  **Daquan Zhou**[5]  **Song Han**[1 2]

**github.com/NVlabs/Sana**

## Abstract

This paper presents SANA-1.5, a linear Diffusion Transformer for efficient scaling in text-to-image generation. Building upon SANA-1.0, we introduce three key innovations: (1) Efficient Training Scaling: A depth-growth paradigm that enables scaling from 1.6B to 4.8B parameters with significantly reduced computational resources, combined with a memory-efficient 8-bit optimizer. (2) Model Depth Pruning: A block importance analysis technique for efficient model compression to arbitrary sizes with minimal quality loss. (3) Inference-time Scaling: A repeated sampling strategy that trades computation for model capacity, enabling smaller models to match larger model quality at inference time. Through these strategies, SANA-1.5 achieves a text-image alignment score of 0.72 on GenEval, which can be further improved to 0.80 through inference scaling, establishing a new SoTA on GenEval benchmark. These innovations enable efficient model scaling across different compute budgets while maintaining high quality, making high-quality image generation more accessible. Our code and pre-trained models will be released.

## 1. Introduction

Text-to-image diffusion models have demonstrated remarkable progress in the past year, with a clear trend towards larger model sizes. Although scaling up the size of the model has proven effective in improving the quality of generation, it comes with substantial computational costs. For instance, recent industry models have grown from PixArt's 0.6B parameters (Chen et al., 2024b) to 24B in Playground v3 (Liu et al., 2024a), resulting in prohibitive training and inference costs for most practitioners.

In contrast, SANA-1.0 (Xie et al., 2024) introduced an efficient linear diffusion transformer that achieved competitive performance while significantly reducing computational requirements. Building upon this foundation, this work explores two fundamental questions: *i) how is the scalability of linear diffusion transformer; ii) how can we scale up large linear DiT and reduce the training cost?*

This paper presents SANA-1.5, which introduces three key innovations for efficient model scaling in both training and inference time. First, we propose an efficient model growth strategy that enables scaling SANA from 1.6B (20 blocks) to 4.8B parameters (60 blocks) while reusing the knowledge learned in the smaller model. Unlike traditional scaling approaches that train large models from scratch, our method initializes additional blocks strategically, allowing the large model to retain the prior knowledge of the small model. This approach reduces training time by 60% compared to training from scratch, as shown in Figure 2.

Second, we introduce a model depth pruning technique that enables efficient model compression. By analyzing block importance through input-output similarity patterns in diffusion transformers, we prune less important blocks and quickly recover the model quality through fine-tuning (e.g., 5 minutes on a single GPU). Our grow-then-prune approach effectively compresses the 60-block model to various configurations (40/30/20 blocks) while maintaining competitive quality, providing an efficient path for flexible model deployment across different compute budgets.

Third, we propose an inference-time scaling strategy for SANA, which enables smaller models to match larger model quality through compute rather than parameter scaling. By generating multiple samples and leveraging a VLM-based selection mechanism, our approach improves the GenEval score from 0.81 to 0.96. This improvement follows a similar log-linear scaling pattern observed in LLMs (Brown et al.,

---

[*]Equal contribution  [1]NVIDIA [2]MIT [3]Tsinghua University [4]Playground [5]Peking University. Correspondence to: Enze Xie <enzex@nvidia.com>, Song Han <songhan@mit.edu>.

*Proceedings of the 42$^{nd}$ International Conference on Machine Learning*, Vancouver, Canada. PMLR 267, 2025. Copyright 2025 by the author(s).

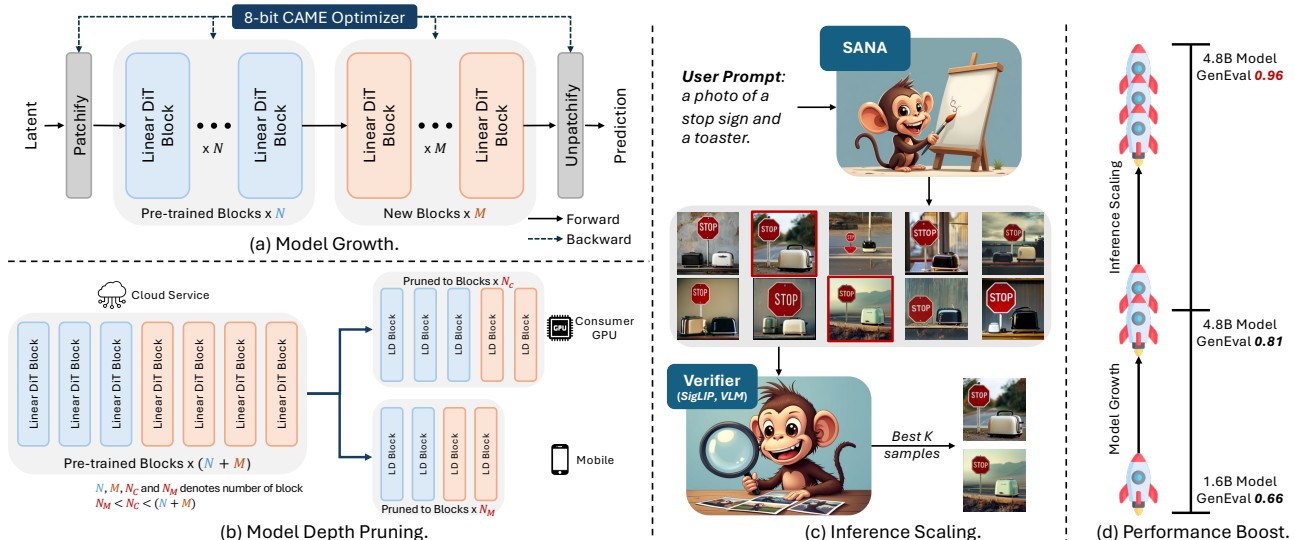

*Figure 1.* **The overall framework of SANA-1.5.** (a) Model Growth: We initialize the large model with a pre-trained small model, and train the large model with 8-bit CAME, which largely accelerates the training convergence and reduces VRAM requirements. (b) Model Pruning: After training the large model, smaller models of different sizes are pruned and fine-tuned for different situations. (c-d) Inference Scaling: We repeat generating many samples with SANA and use VLM as a verifier to select the best-of-N samples, which largely boosts the quality.

2024), demonstrating that computational resources can be effectively traded for model capacity, challenging the conventional wisdom that larger models are always necessary for better quality.

rThese three technical contributions - model growth, model depth pruning and inference scaling - form a coherent framework for efficient model scaling. The model growth strategy first explores a larger optimization space, discovering better feature representations. The model depth pruning then identifies and preserves these essential features, enabling efficient deployment. Meanwhile, inference-time scaling provides a complementary perspective. When model capacity is constrained, we can utilize extra inference-time computational resources to achieve similar or even better results than larger models. Together, these approaches demonstrate that thoughtful optimization strategies can outperform simple parameter scaling, providing multiple paths to achieve high quality under different resource constraints.

To enable efficient training and fine-tuning large models, we implement a memory-efficient optimizer CAME-8bit by extending CAME (Luo et al., 2023) with block-wise 8-bit quantization (Dettmers et al., 2021). CAME-8bit reduces memory usage by $\sim 8\times$ compared to AdamW-32bit (Loshchilov, 2017) while maintaining training stability. This optimization proves effective not only in pre-training but is particularly valuable for single-GPU fine-tuning scenarios, enabling researchers to fine-tune SANA-4.8B on consumer GPUs like RTX 4090, making large model fine-tuning more accessible to the open-source community.

Our extensive experiments demonstrate that SANA-1.5 achieves $2.5\times$ faster training convergence than the traditional approach (i.e., scale up and train from scratch). Through our training scaling strategy, we improve the GenEval score from 0.66 to 0.81, which can be further boosted to 0.96 with inference scaling, establishing a new state-of-the-art on the GenEval benchmark. More importantly, our findings reveal a fundamental insight: efficient scaling can be achieved through better optimization trajectories rather than simply increasing model capacity. By leveraging knowledge from smaller models and carefully designing the growth-pruning process, we show that the path to better quailty does not always require larger models.

In summary, SANA-1.5 introduces a new perspective on model scaling in text-to-image generation. Rather than following the conventional paradigm "bigger is better", we demonstrate that the growth and pruning of strategic models, combined with the inference-time scaling, can achieve comparable or better results with significantly reduced training resources. This approach not only advances the theoretical understanding of model scaling but also makes high-quality text-to-image generation more accessible to the broader research community and practical applications.

## 2. Methods

### 2.1. Overview

Increasingly larger models have dominated text-to-image generation, but SANA-1.5 introduces a different paradigm

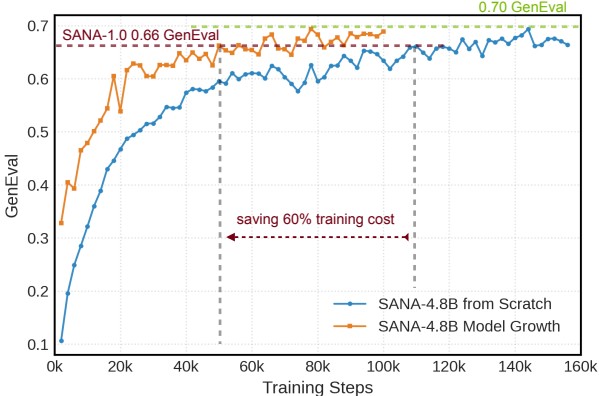

*Figure 2.* **Training efficiency comparison of different initialization strategies.** Training curves on GenEval benchmark for SANA-1.5 4.8B using model growth strategy *vs* training from scratch. Our model growth approach achieves the same performance (0.70 GenEval) with 60% fewer training steps, significantly improving training efficiency.

that achieves efficient scaling through three complementary strategies. Rather than training large models from scratch, we first expand a base model with $N$ transformer layers to $N + M$ layers (where $N = 20$, $M = 40$ in our experiments) while preserving its learned knowledge. During inference, we employ two complementary approaches for efficient deployment: (1) a model depth pruning mechanism that identifies and preserves essential transformer blocks, enabling flexible model configurations with small fine-tuning cost, and (2) an inference scaling strategy that trades computation for model capacity through repeat sampling and VLM-guided selection. Meanwhile, our memory-efficient CAME-8bit optimizer makes it possible to fine-tune billion-scale models on a single consumer GPU. Figure 1 illustrates how these components work together to achieve efficient scaling in different computational budgets.

## 2.2. Efficient Model Growth

Rather than training large models from scratch, we propose an efficient model growth strategy that expands a pre-trained DiT with $N$ layers to $N + M$ layers while preserving its learned knowledge. We explore three initialization strategies to ensure effective knowledge transfer during model expansion. Figure 11 in the appendix illustrates the three strategies.

**Initialization Strategies** Let $\theta_i \in \mathbb{R}^d$ denote the parameters of layer $i$ in the expanded model and $\theta_i^{\text{pre}} \in \mathbb{R}^d$ represent the parameters of layer $i$ of the pre-trained model, where $d$ is the parameter dimension of each layer. We investigate three approaches for parameter initialization:

(1) Partial Preservation Init, where we preserve the first $N$ pre-trained layers and randomly initialize the additional $M$

layers, with special handling of key components. Formally, for $i$-th layer index:

$$\theta_i = \begin{cases} \theta_i^{\text{pre}}, & \text{if } i < N \\ \mathcal{N}(0, \sigma^2), & \text{if } i \geq N \end{cases},$$

where $\mathcal{N}(0, \sigma^2)$ is the normal distribution with standard deviation as $\sigma$.

(2) Cyclic Replication Init, which repeats the pre-trained layers periodically. For $i$-th layer in the expanded model:

$$\theta_i = \theta_{i \bmod N}^{\text{pre}}$$

(3) Block Replication Init, which extends each pre-trained layer into consecutive layers. Given expansion ratio $r = M/N$, for pre-trained $i$-th layer, it initializes $r$ consecutive layers in the expanded model:

$$\theta_{ri+j} = \theta_i^{\text{pre}}, \quad \text{for } j \in \{0, \ldots, r-1\}, i \in [0, N-1],$$

where $r$ represents the expansion ratio (*e.g.*, $r = 3$ when expanding from 20 to 60 layers), $\theta_i$ denotes the parameters of layer $i$ in the expanded model, $\theta_i^{\text{pre}}$ represents the parameters from the pre-trained model.

**Stability Enhancement** To ensure training stability across all initialization strategies, we incorporate layer normalization for query and key components in both linear self-attention and cross-attention modules. This normalization technique is crucial as it: (1) stabilizes the attention computation during the early stages of training, (2) prevents potential gradient instability when integrating new layers, and (3) enables rapid adaptation while maintaining model quality.

**Identity Mapping Initialization** We initialize the weights of specific components to zero in new layers, particularly the output projections of self-attention, cross-attention, and the final point-wise convolution in MLP blocks, following (Chen et al., 2015). This zero-initialization ensures that new transformer blocks initially behave as identity functions, providing two key benefits: (1) exact preservation of the pre-trained model's behavior at the start of training, and (2) stable optimization path from a known good solution.

**Design Choice** Among these strategies, we adopt the partial preservation initialization approach for its simplicity and stability. This choice creates a natural division of labor: the pre-trained $N$ layers maintain their feature extraction capabilities while the randomly initialized $M$ layers, starting from identity mappings, gradually learn to refine these representations. Empirically, this approach provides the most stable training dynamics compared to

cyclic and block expansion strategies. Considering the block importance (see analysis in Section 3.3), we drop the last two blocks in the pre-trained model to enhance the learning of newly added blocks.

## 2.3. Memory-Efficient CAME-8bit Optimizer

Building upon CAME (Luo et al., 2023) and AdamW-8bit (Dettmers et al., 2021), we propose CAME-8bit for efficient large-scale model training. CAME reduces memory usage by half compared to AdamW through matrix factorization of second-order moments, making it particularly efficient for large linear and convolutional layers. We further extend CAME with block-wise 8-bit quantization for first-order moments, while preserving 32-bit precision for critical statistics to maintain optimization stability. This hybrid approach reduces the optimizer's memory footprint to approximately 1/8 of AdamW, enabling billion-scale model training on consumer GPUs without compromising convergence properties.

**Block-wise Quantization Strategy** We adopt a selective quantization approach where only large matrices (>16K parameters) in linear and 1×1 convolution layers are quantized, as these layers dominate the optimizer's memory footprint. For each block of size 2048, we compute independent scaling factors to preserve local statistical properties. Given a tensor block $x \in \mathbb{R}^n$ representing the first-order momentum values, the quantization function $q(x)$ maps each value to an 8-bit integer:

$$q(x) = \text{round}\left(\frac{x - \min(x)}{\max(x) - \min(x)} \times 255\right), \quad (1)$$

where $\min(x)$ and $\max(x)$ are the minimum and maximum values in the block respectively, and $\text{round}(\cdot)$ maps to the nearest integer. This linear quantization preserves the relative magnitude of values within each block while compressing the storage to 8 bits per value.

**Hybrid Precision Design** To maintain optimization stability, we keep second-order statistics in 32-bit precision, as these are critical for proper gradient scaling. Benefiting from CAME's matrix factorization, these statistics are already memory-efficient: for a linear layer with $d_{in}$ input dimensions and $d_{out}$ output dimensions, the storage of second-order moments is reduced from $O(d_{in} \times d_{out})$ to $O(d_{in}+d_{out})$, making their precision less critical for overall memory consumption. This hybrid approach reduces memory usage while preserving CAME's convergence properties. Memory reduction can be formulated as:

$$M_{\text{saved}} = \sum_{l \in \mathcal{L}} (n_l \times 24) \text{ bytes}, \quad (2)$$

where $\mathcal{L}$ is the set of quantized layers, $n_l$ is the parameter count of layer $l$, and 24 represents the maximum bytes saved per parameter. In practice, the actual memory savings are slightly lower due to several factors: (1) small layers (<16K parameters) remain in 32-bit precision, (2) second-order statistics are kept in 32-bit, and (3) additional overhead from quantization metadata. Nevertheless, this approximation provides a good estimate of the memory efficiency gained through our hybrid quantization strategy.

## 2.4. Model Depth Pruning

To address the challenge of balancing effectiveness and efficiency in large models, we introduce a model depth pruning approach that efficiently compresses large models into various smaller configurations while maintaining comparable quality. Inspired by Minitron (Sreenivas et al., 2024), a transformer compression technique for LLMs, we analyze block importance through input-output similarity patterns:

$$\text{BI}_i = 1 - \mathbb{E}_{X,t} \frac{\mathbf{X}_{i,t}^T \mathbf{X}_{i+1,t}}{\|\mathbf{X}_{i,t}\|_2 \|\mathbf{X}_{i+1,t}\|_2}, \quad (3)$$

where $\mathbf{X}_{i,t}$ denotes the input of the $i$-th transformer block. We average the block importance across diffusion time-steps and our calibration dataset, which contains 100 diverse prompts. As shown in Figure 5c, the block importance is higher in head and tail blocks, and we conjecture that the head blocks change the latent distribution to diffusion distribution and the tail blocks change it back. The middle blocks commonly have higher similarity between input and output features, demonstrating the gradual refinement of the generated results. We prune the transformer blocks based on the importance of the sorted block. As illustrated in Figure 4, pruning the blocks will gradually impair the high-frequency details. Therefore, after pruning, we further fine-tune the model to compensate for the information loss. Specifically, we use the same training loss as the large model to supervise the pruned models. Adapting the pruned model to complete information is surprisingly easy. With only 100 fine-tune steps, the pruned 1.6B model can achieve comparable quality with the full 4.8B model and outperform the SANA-1.0 1.6B model (Table 3).

## 2.5. Inference-Time Scaling

With sufficient training, SANA-1.5 gains stronger generation abilities after efficient model growth. Inspired by the recent success of inference-time scaling in large language models (LLMs) (Brown et al., 2024), we are interested in inference-time scaling to push the generation upper bound.

**Scaling Denoising Steps v.s. Scaling Samplings** For SANA and many other diffusion models, a natural option to scale up the inference-time computation is to increase the

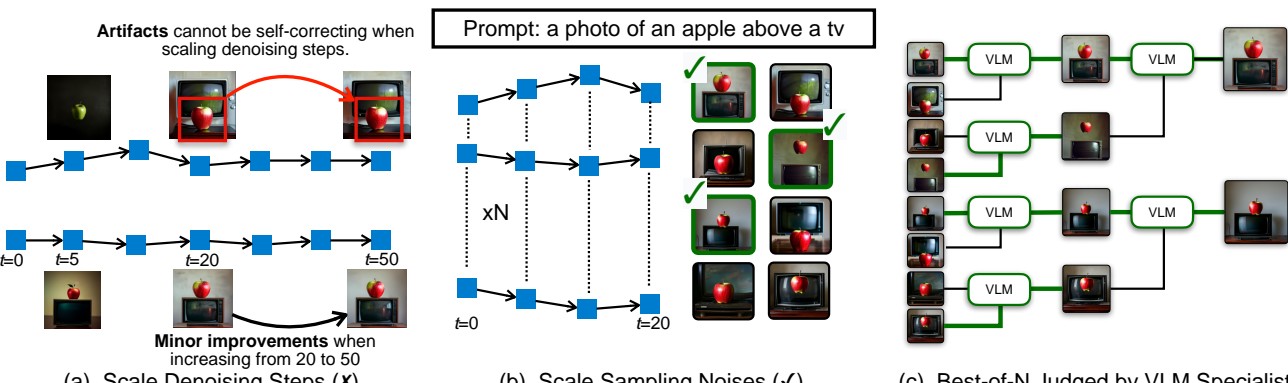

*Figure 3.* **Comparing Scaling Between Denoising Steps and Samples with VLM Judgment Visualization**. (a) Scaling denoising steps show only minor improvements and often fail to self-correct artifacts, making it a poor option for scaling up. (b) In contrast, scaling sampling noise proves more effective, with VLM specialists helping to verify and select images that match the prompts. (c) VLM evaluates and ranks the best images in a tournament format.

number of denoising steps. However, using more denoising steps is not ideal for scaling for two reasons. First, additional denoising steps cannot self-correct errors. Figure 3(a) illustrates this with a sample, where objects misplaced at an early stage remained unchanged in subsequent steps. Second, the generation quality quickly reaches a plateau. As shown in Figure 3, SANA produces visually pleasing results with just 20 steps, showing no significant visual improvement even increase 2.5× steps.

In contrast, scaling the number of sampling candidates is a more promising direction. As presented in Figure 3(b), a small model SANA (1.6B) can also generate correct results for difficult test prompts when given multiple attempts, much like a sloppy/scattered student who can draw as requested but sometimes makes mistakes during execution. With enough opportunities to try, it can still provide a satisfactory answer. Therefore, we choose to generate more images and introduce a "patient teacher" to score the results, which will be expanded in the following.

**Visual Language Model (VLM) as the Judge**   To find images that best match a given prompt, a model that understands both text and images is needed. While popular models like CLIP (Radford et al., 2021) and SigLIP (Zhai et al., 2023) offer multi-modal capabilities, their small context windows (77 tokens for CLIP and 66 tokens for SigLIP) limit their effectiveness. This limitation poses a problem since SANA usually takes long, detailed descriptions as inputs. To address this, we explored Visual Language Models to evaluate prompt-matching for generated images. We tested commercial multi-modal APIs, specifically GPT-4o (OpenAI, 2023b) and Gemini-1.5-pro (Team, 2024), but encountered two significant issues. First, when evaluating single images against prompts, both APIs lacked consistency in their ratings across different runs. Second, when tasked with select-

ing the best-matching images from multiple options, both models exhibited a strong bias toward the first-presented options, regardless of image ordering or shuffling.

Instead of applying existing models or APIs, we trained a specialized NVILA-2B (Liu et al., 2024b) to score the images, which we named **SaVILA**: Sana generates then VILA picks. We created a 2M prompt-matching dataset with prompts generated in the GenEval style. We excluded the prompts that are already existed in GenEval evalset and generated the images using using Flux-Schnell (Labs, 2024) to avoid overfitting. We then formatted these as multimodal conversations, as shown below

- ```
  User:  You are an AI assistant
  specializing in image analysis and
  ranking.  Your task is to analyze and
  compare image based on how well they
  match the given prompt.  <image> The
  given prompt is:  <prompt>.  Please
  consider the prompt and the image to
  make a decision and response directly
  with 'yes' or 'no'
  ```

- ```
  SaVILA: 'yes' / 'no'.
  ```

The fine-tuned SaVILA can effectively assess how well images match their prompts and robustly filters out prompt-mismatching images. During inference, we compare two images in each round until the top-N candidates are determined

- When SaVILA responses one 'yes' and one 'no', we pick the image with 'yes';

- When SaVILA responses both 'yes' or 'no', we pick the image with higher confidence (logprob);

Such tournament-style comparison robustly filters out prompt-mismatching images and consistently boosts the GenEval scores, as illustrated in Figure 3(c) and Table. 2. More details of inference time scaling are attached in Appendix B.

## 3. Experiments

### 3.1. Experimental Setup

**Model Architecture.** Our final model (SANA-4.8B) scales to 60 layers while maintaining the same channel dimension (2240 per layer) and FFN dimension (5600) as SANA-1.6B. The architecture, training data, and other hyperparameters remain consistent with SANA-1.6B (Xie et al., 2024).

**Training Details.** We conduct distributed training using PyTorch DDP across 64 NVIDIA A100 GPUs on 8 DGX nodes. Our training pipeline follows a two-phase strategy: we first pre-train the model with a learning rate of 1e-4, followed by supervised fine-tuning (SFT) with a reduced learning rate of 2e-5. The global batch size is dynamically adjusted between 1024 and 4096 throughout the training process. Following common practice in large language model training, we initially pre-train on a large-scale dataset before performing SFT on a high-quality dataset.

**Evaluation Protocol.** We adopt multiple evaluation metrics, including FID, CLIP Score, GenEval (Ghosh et al., 2024), and DPG Bench (Hu et al., 2024b), comparing it with SOTA methods. FID and Clip Score are evaluated on the MJHQ-30K (Li et al., 2024a) dataset, which contains 30K images from Midjourney. GenEval and DPG-Bench both focus on measuring text-image alignment, with 553 and 1,065 test prompts, respectively. We particularly emphasize the GenEval as it better reflects text-image alignment and shows more room for improvement than other metrics.

### 3.2. Main Results

**Model Growth** We compare SANA-4.8B with the most advanced text-to-image generation methods in Table 1. The scaling from SANA-1.6B to 4.8B Pre (Pre-trained) brings substantial improvements: 0.06 absolute gains in GenEval (from 0.66 to 0.72), 0.34 reduction in FID (from 5.76 to 5.42), and 0.2 improvement in DPG score (from 84.8 to 85.0). Compared to state-of-the-art methods, our 4.8B model achieves comparable or better results than much larger models like Playground v3 (24B) and FLUX (12B) while using only a fraction of their parameters. We further perform post-training on the 4.8B model using a high-quality dataset, with details provided in Sec. 3.3. To differentiate between the models, we refer to the pre-trained model as Pre and the post-trained model as Ours. After post-training, SANA-4.8B achieves a significant performance improvement on the GenEval benchmark, increasing from 0.72

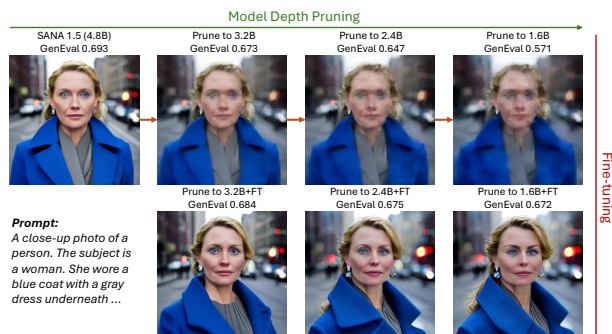

*Figure 4.* **Visual comparison of SANA-1.5 models with different pruned configurations.** Our adaptive depth pruning enables efficient compression to various model sizes (from 1.6B to 4.8B). While aggressive pruning may slightly affect fine-grained details, the semantic content is well preserved, and the overall image quality can be easily recovered after brief fine-tuning (100 steps on 1 GPU), demonstrating the effectiveness of our pruning strategy.

to 0.81. Notably, SANA-4.8B demonstrates 0.81 GenEval score, outperforming Playground v3's 0.76, but with 5.5 times lower latency than FLUX-dev (23.0s). Our model also maintains 6.5 times higher throughput than these larger models (compared to FLUX-dev's 0.04 samples/s), making it more practical for real-world applications. The speed is tested on one A100 GPU with FP16 Precision.

**Model Pruning** We compare among difference sizes of SANA-1.5 and SANA-1.0 models in Figure 4 and Table 3. For a fair comparison with SANA-1.0 1.6B, the SANA-1.5 4.8B model here is trained without supervised fine-tuning from high-quality data. All results are evaluated on images of size 512×512. With a small computational cost, the pruned and fine-tuned model outperforms the model trained from scratch (0.672 *v.s.* 0.664), which is an efficient approach to obtaining models of various sizes.

**Inference Scaling** We incorporate inference scaling with the SANA-1.5 4.8B v2 model and compare it against other large image generation models on the GenEval benchmark (Table 2). By selecting samples from 2048 generated images, the inference-scaled model outperforms naive single-image generation by 15% in overall accuracy from 0.81 to 0.96, with particularly significant improvements in the "Position" (from 0.59 to 0.96), and "Color Attribution" (from 0.65 to 0.87). Furthermore, equipped with inference scaling, our 4.8B model outperforms Playground v3 (24B) by 20% in overall accuracy (0.76 vs 0.96). These results demonstrate that trading inference efficiency can enhance model generation quality and accuracy, even with much smaller base model size (4.8B vs 24B).

*Table 1.* **Comprehensive comparison of our method with SOTA approaches in efficiency and performance.** The speed is tested on one A100 GPU with BF16 Precision. Throughput: Measured with batch=10. Latency: Measured with batch=1 and sampling step=20. We highlight the **best** and second best entries.

| Methods | Throughput (samples/s) | Latency (s) | Params (B) | FID ↓ | CLIP ↑ | GenEval ↑ | DPG ↑ |
|---|---|---|---|---|---|---|---|
| LUMINA-Next (Zhuo et al., 2024) | 0.12 | 9.1 | 2.0 | 7.58 | 26.84 | 0.46 | 74.6 |
| SDXL (Podell et al., 2023) | 0.15 | 6.5 | 2.6 | 6.63 | 29.03 | 0.55 | 74.7 |
| Playground v2.5 (Li et al., 2024a) | 0.21 | 5.3 | 2.6 | 6.09 | 29.13 | 0.56 | 75.5 |
| Hunyuan-DiT (Li et al., 2024d) | 0.05 | 18.2 | 1.5 | 6.54 | 28.19 | 0.63 | 78.9 |
| PixArt-Σ (Chen et al., 2024b) | 0.4 | 2.7 | 0.6 | 6.15 | 28.26 | 0.54 | 80.5 |
| DALLE 3 (OpenAI, 2023a) | - | - | - | - | - | 0.67 | 83.5 |
| SD3-medium (Esser et al., 2024) | 0.28 | 4.4 | 2.0 | 11.92 | 27.83 | 0.62 | 84.1 |
| FLUX-dev (Labs, 2024) | 0.04 | 23.0 | 12.0 | 10.15 | 27.47 | 0.67 | 84.0 |
| FLUX-schnell (Labs, 2024) | 0.5 | 2.1 | 12.0 | 7.94 | 28.14 | 0.71 | 84.8 |
| Playground v3 (Liu et al., 2024a) | 0.06 | 15.0 | 24 | - | - | 0.76 | **87.0** |
| SANA-1.0 0.6B (Xie et al., 2024) | 1.7 | 0.9 | 0.6 | 5.81 | 28.36 | 0.64 | 83.6 |
| SANA-1.0 1.6B (Xie et al., 2024) | 1.0 | 1.2 | 1.6 | 5.76 | 28.67 | 0.66 | 84.8 |
| **SANA-1.5 4.8B Pre** | 0.26 | 4.2 | 4.8 | **5.42** | 29.16 | 0.72 | 85.0 |
| **SANA-1.5 4.8B Ours** | 0.26 | 4.2 | 4.8 | 5.99 | **29.23** | **0.81** | 84.7 |

*Table 2.* **Detailed GenEval evaluation benchmark.** SANA-1.5 + Inference Scaling with 2048 samples achieves absolute SoTA compared to open-source and commercial methods. We used the numbers from Playground v3 (Liu et al., 2024a) for the baseline methods.

| Method | Overall | Single | Two | Counting | Colors | Position | Color Attribution |
|---|---|---|---|---|---|---|---|
| SDXL (Podell et al., 2023) | 0.55 | 0.98 | 0.74 | 0.39 | 0.85 | 0.15 | 0.23 |
| DALLE 3 (OpenAI, 2023a) | 0.67 | 0.96 | 0.87 | 0.47 | 0.83 | 0.43 | 0.45 |
| Flux-dev (Labs, 2024) | 0.68 | 0.99 | 0.85 | 0.74 | 0.79 | 0.21 | 0.48 |
| SD3 (Esser et al., 2024) | 0.74 | 0.99 | 0.94 | 0.72 | 0.89 | 0.33 | 0.60 |
| Playground v3 (Liu et al., 2024a) | 0.76 | 0.99 | 0.95 | 0.72 | 0.82 | 0.50 | 0.54 |
| SD1.5 (Rombach et al., 2022a) | 0.42 | 0.98 | 0.39 | 0.31 | 0.72 | 0.04 | 0.06 |
| + Inference Scaling | **0.87** | **1.00** | **0.97** | **0.93** | **0.96** | **0.75** | **0.62** |
| SANA-1.5 4.8B Pre | 0.72 | 0.99 | 0.85 | 0.77 | 0.87 | 0.34 | 0.54 |
| SANA-1.5 4.8B Ours | 0.81 | 0.99 | 0.93 | 0.86 | 0.84 | 0.59 | 0.65 |
| + Inference Scaling | **0.96** | **1.00** | **1.00** | **0.97** | **0.94** | **0.96** | **0.87** |

*Table 3.* **Evaluation of pruned SANA models.** "3.2B" and "1.6B" denote the model directly pruned from SANA-1.5 4.8B, and "+FT" denotes efficiently fine-tuning the pruned model.

| Method | 4.8B | 3.2B | +FT | 1.6B | +FT | SANA-1.0 1.6B |
|---|---|---|---|---|---|---|
| GenEval ↑ | 0.693 | 0.673 | 0.684 | 0.571 | 0.672 | 0.664 |

*Table 4.* **Model performance across different scales.** Models are first pre-trained and then fine-tuned (SFT) on high-quality data.

| Params. (B) | Stage | Train Steps | GenEval ↑ |
|---|---|---|---|
| 0.6 | pre-train | >200K | 0.64 |
|  | + SFT | ~10K | 0.68 (+4%) |
| 1.6 | pre-train | >200K | 0.66 |
|  | + SFT | ~10K | 0.69 (+3%) |
| 4.8 | pre-train | >100K | 0.69 |
|  | + SFT | ~10K | 0.72 (+3%) |

## 3.3. Analysis

**Comparison of Different Optimizers** We compare CAME-8bit with AdamW-8bit and their 32-bit counterparts in Figure 6 for SANA-1.6B training. The 8-bit optimizers (AdamW-8bit, Came-8bit) achieve comparable convergence to their 32-bit counterparts while significantly reducing GPU memory usage. Specifically, CAME-8bit reduces memory consumption by 25% compared to AdamW (43GB vs 57GB)

with no degradation in training convergence speed. Note that CAME-8bit reduces optimizer state memory usage proportionally to model size, yielding greater memory savings for larger models.

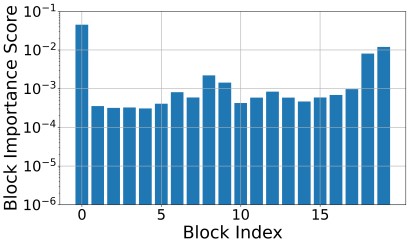 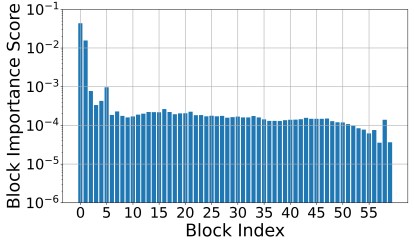 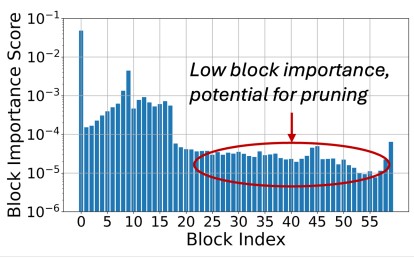

(a) SANA 1.6B trained from scratch.  (b) SANA-4.8B trained from scratch.  (c) SANA 4.8B with model growth.

*Figure 5.* **Analysis of block importance (BI) across different models**: (a) SANA-1.0 1.6B, (b) SANA 4.8B trained from scratch, and (c) our final SANA-1.5 4.8B with initialization.

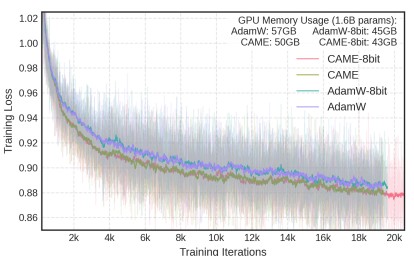 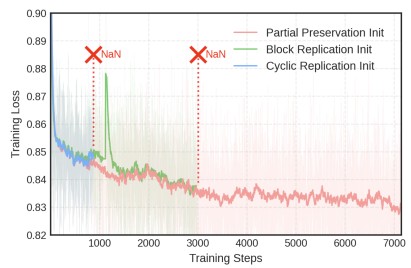 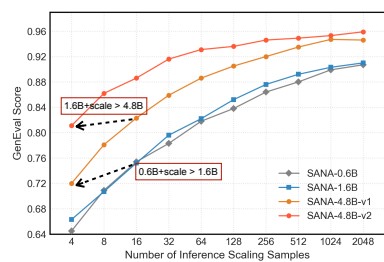

*Figure 6.* **Training loss curves** for different optimizers on a 1.6B parameter diffusion model. The CAME-8bit reduces memory consumption by 25% compared to AdamW while maintaining the convergence speed.

*Figure 7.* **Comparison of different initialization strategies.** Partial Preservation Init shows stable training behavior while Cyclic and Block Replication strategies suffer from training instability (NaN losses).

*Figure 8.* **Inference-time Scaling Results**. Scaling up inference time compute consistently yields better GenEval scores, and helps the small model to achieve comparable or better performance with larger ones.

**Comparison of Different Initialization Strategies** We compare the three types of initialization strategies in Figure 7. Partial Preservation Init shows stable training behavior while Cyclic and Block Replication strategies suffer from training instability (NaN losses). Such observation is also supported by the block importance analysis in Figure 5. The feature distribution of the 4.8B model is different from the 1.6B model due to the model capacity, and thus, replication of the block weight increases the difficulty of convergence to the final distribution.

**Block Importance Instructing Model Growth** The analysis of block importance instructs both our initialization strategy and pruning approach. The block importance of the pre-trained SANA-1.0 model is shown in Figure 5a, where more information resides in the head and tail blocks. During model scaling, we initially attempted to append new blocks directly after all the pre-trained blocks. However, we observed that the newly added blocks failed to learn effective information and became stuck in local minima. The primary reason is that the well-learned pre-trained features dominate the feature representation through skip connections. Therefore, we remove the last two blocks, which are more task-relevant, before adding new blocks. This process effectively facilitates learning in the later blocks.

**Block Importance Instructing Model Depth Pruning** As shown in Figure 5c, blocks in the middle to the end have low importance scores, especially when compared with the model trained from scratch (Figure 5b). This indicates potential for model size reduction. Based on this observation, we prune the blocks in SANA-1.5 4.8B according to their sorted importance scores. In Figure 4, pruning the blocks (gradually reducing from 60 to 20 blocks) impairs high-frequency information. The lack of high-frequency details degrades the accuracy of GenEval benchmark to 0.571. However, the image layout and semantic information are well preserved. Therefore, high-frequency information can be quickly recovered with 100 steps of fine-tuning on a single GPU.

**High-quality Data Fine-tuning** While extensive pre-training on large-scale datasets leads to quality saturation, fine-tuning on a curated dataset (3M samples from 50M pre-training data) significantly and efficiently improves model capabilities of different model sizes. Specifically, by fine-tuning on image-text pairs with CLIP score > 25, our 4.8B model achieves a 3% improvement in GenEval score compared to the pre-trained model, as shown in Table 4.

Furthermore, we created a new small-scale SFT data set, consisting of 144,291 images generated from 18,240

prompts in the GenEval style. Each prompt generates 7.91 "correct" images on average. We limit each prompt to generate at most 10 correct images to prevent overfitting. The correct image is filtered by the GenEval toolkit. Note that we excluded the prompts already existing in GenEval evalset and generated images using Flux-Schnell to avoid overfitting. As shown in Table 2, this new SFT data improves the GenEval result of SANA-1.5-4.8B from 0.72 (v1) to 0.81 (v2). Figure 9 presents a comprehensive comparison between SANA-1.5-4.8B and current state-of-the-art methods across multiple challenging scenarios. The results demonstrate SANA-1.5-4.8B's consistent superiority in generated image quality, as evidenced by both quantitative metrics and visual assessment.

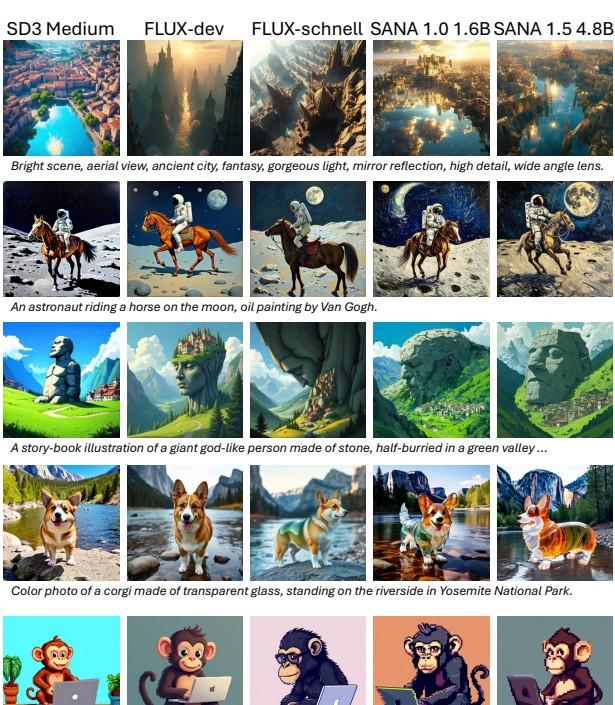

SD3 Medium  FLUX-dev  FLUX-schnell  SANA 1.0 1.6B  SANA 1.5 4.8B

*Bright scene, aerial view, ancient city, fantasy, gorgeous light, mirror reflection, high detail, wide angle lens.*

*An astronaut riding a horse on the moon, oil painting by Van Gogh.*

*A story-book illustration of a giant god-like person made of stone, half-burried in a green valley ...*

*Color photo of a corgi made of transparent glass, standing on the riverside in Yosemite National Park.*

*A smart monkey with a laptop, in pixel art style*

*Figure 9.* **Comparison of SANA-1.5 against SoTA methods.** We evaluate SANA-1.5 alongside contemporary approaches using identical text prompts, including challenging cases with noisy inputs and long-form text. For detailed assessment, please zoom into the comparative samples.

**Inference Time Scaling**  Figure 8 demonstrates the benefits of scaling up inference-time computation. First, SANA's accuracy on GenEval consistently improves with more samplings. Second, inference-time scaling enables smaller SANA models to match or even surpass the accuracy of larger ones (1.6B + scaling is better than 4.8B). This reveals the potential of scaling up inference and allows SANA to push toward new state-of-the-art results. As shown in Table 2, our best SANA model with inference scaling outperforms all previous commercial and community models.

The only limitation is increased computational cost: sampling $N$ images requires $N \times 49,140G$ FLOPs for SANA generation and $2N \times 4,518G$ FLOPs for SaVILA judgment and comparison. We leave the efficiency for future work.

## 4. Related Work

We put a relatively brief overview of related work in the main text, with a more comprehensive version in the appendix. Text-to-image generation has evolved rapidly, from Stable Diffusion (Rombach et al., 2022b) to more recent architectures like DiT (Peebles & Xie, 2022) and its variants (Chen et al., 2024c; Labs, 2024; Esser et al., 2024). Efficiency-focused works like SnapGen (Hu et al., 2024a), PixArt-$\alpha$ (Chen et al., 2024c) and SANA (Xie et al., 2024) have significantly reduced training and inference costs. Autoregressive models (Tang et al., 2024; Tian et al., 2024; Sun et al., 2024) are also developed rapidly and achieve comparable quality as diffusion models. Research in language (Kaplan et al., 2020) and vision domains (Li et al., 2024b; Liang et al., 2024) both revealed power-law relationships. In the image generation field, (Xu et al., 2023; 2024) has explored RLHF to align model with human preferences. More recent concurrent work (Singhal et al., 2025; Ma et al., 2025) have explored various strategies to improve generation quality without increasing model size.

## 5. Conclusion

This paper presents a comprehensive approach to efficient model scaling, addressing both training and inference compute challenges. For training efficiency, we propose a memory-efficient optimizer CAME-8bit and a stable model growth strategy. For inference scaling and acceleration, we introduce repeat sampling and depth pruning techniques. These approaches collectively enable significant quality improvements under limited computing budgets, making large-scale generative models more accessible. This work contributes to democratizing large-scale AI research by making it more accessible to researchers with limited resources.

## Impact Statement

Misusing generative AI models to generate NSFW content is a challenging issue for the community. To enhance safety, we have equipped SANA together with a safety check model (Zeng et al., 2024a). Specifically, the user prompt will first be sent to the safety check model to determine whether it contains NSFW(not safe for work) content. While SANA-1.5 demonstrates efficient model scaling, challenges remain in complex generation tasks, particularly text rendering and human details. Our work makes large-scale model training more accessible, while encouraging responsible development and deployment to prevent misuse.

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

# A. Full Related Work

**Text to Image Generation** Text-to-image generation has undergone rapid evolution in model architectures and efficiency. The field gained momentum with Stable Diffusion (Rombach et al., 2022b), and later witnessed a pivotal shift towards Diffusion Transformers (DiT) (Peebles & Xie, 2022) architectures. PixArt-$\alpha$ (Chen et al., 2024c) demonstrated competitive quality while significantly reducing training costs to just 10.8% of Stable Diffusion v1.5's requirements (Rombach et al., 2022b). Recent large-scale models like FLUX (Labs, 2024) and Stable Diffusion 3 (Esser et al., 2024) have pushed the boundaries in compositional generation capabilities, while Playground v3 (Liu et al., 2024a) achieved state-of-the-art image quality through full integration with Large Language Models (LLMs) (Dubey et al., 2024). PixArt-$\Sigma$ (Chen et al., 2024b) further enabled direct 4K resolution image generation with a compact 0.6B parameter model. In parallel, efficiency-focused innovations like SANA (Xie et al., 2024) introduced breakthrough capabilities in high-resolution synthesis through deep compression autoencoding (Chen et al., 2024a) and linear attention mechanisms, making deployment possible even on laptop GPUs. These developments showcase the field's progression toward both more powerful and more accessible text-to-image generation.

**Diffusion Model Pruning** Neural network pruning (Han et al., 2016) is an effective technique for improving the efficiency of neural models, particularly for deployment on resource-constrained devices. By removing redundant weights, it reduces both model size and computational complexity. In LLMs, researchers have successfully applied pruning to shrink models for various applications (Sreenivas et al., 2024; Ma et al., 2023). For generative models, (Li et al., 2020) employ neural architecture search (Cai et al., 2019) to prune GAN channels (Goodfellow et al., 2014). SnapFusion (Li et al., 2024c) extends this to diffusion models, using elastic depth (Cai et al., 2019) to prune UNet blocks (Ho et al., 2020), managing to deploy Stable Diffusion on mobile phones. Similarly, MobileDiffusion (Zhao et al., 2023) shrinks UNet depth and distills the model for single-step inference. Our approach targets the recent DiT architecture (Peebles & Xie, 2023). We instead use a heuristic method to identify and prune less important blocks directly, avoiding the overhead of search.

**Training Scaling in LLM and DiT** Training scaling laws have been extensively studied in both language (Kaplan et al., 2020; Alabdulmohsin et al., 2022) and vision (Li et al., 2024b; Zhai et al., 2021; Liang et al., 2024) domains. For language models, research has revealed power-law relationships between model accuracy and factors like model size, dataset size, and compute (Kaplan et al., 2020). These scaling patterns have been consistently observed across several orders of magnitude. Recently, similar scaling properties have been discovered in diffusion-based text-to-image generation. Studies show that DiT's pre-training loss follows power-law relationships with computational resources (Liang et al., 2024). Furthermore, extensive experiments on scaling both denoising backbones and training sets reveal that increasing transformer blocks is more parameter-efficient than increasing channel numbers for improving text-image alignment. The quality and diversity of the training set prove more crucial than mere dataset size (Li et al., 2024b). These findings provide valuable insights for determining optimal model architectures and data requirements in both domains.

**Inference Scaling Law** Recent studies have revealed significant insights into inference scaling laws for large language models. The pioneering work "Large Language Monkeys" (Brown et al., 2024) discovered that coverage (the fraction of problems solved) scales with the number of samples following a log-linear relationship. Building upon this, self-consistency approaches demonstrated that sampling multiple reasoning paths and selecting the most consistent answer can substantially improve model accuracy (Wang et al., 2022). This was further enhanced by progressive-hint prompting techniques (Zheng et al., 2023), achieving significant gains on various reasoning benchmarks. Recent theoretical work (Wu et al., 2024) shows that smaller models paired with advanced inference algorithms can outperform larger models under the same computation budget. However, studies on compound inference systems (Chen et al., 2024d) reveal that increasing LLM calls shows non-monotonic behavior, performing better on "easy" queries but worse on "hard" ones. These findings collectively demonstrate the importance of optimizing inference strategies rather than simply scaling up model size or increasing the sampling budget. Concurrent works (Singhal et al., 2025; Ma et al., 2025) have also independently explored and validated the effectiveness of inference scaling in diffusion models.

# B. Inference-Time Scaling Details

**Dataset.** To finetune NVILA (Liu et al., 2024b) for SANA inference-time scaling, we generated 2.5M images and evaluated their alignment with the given prompts using the GenEval toolkit. We utilize 15,654 unique prompts and generate 160 images per prompt using Flux-Schnell. Our prompts are constructed in a style similar to GenEval, incorporating 80 object classes

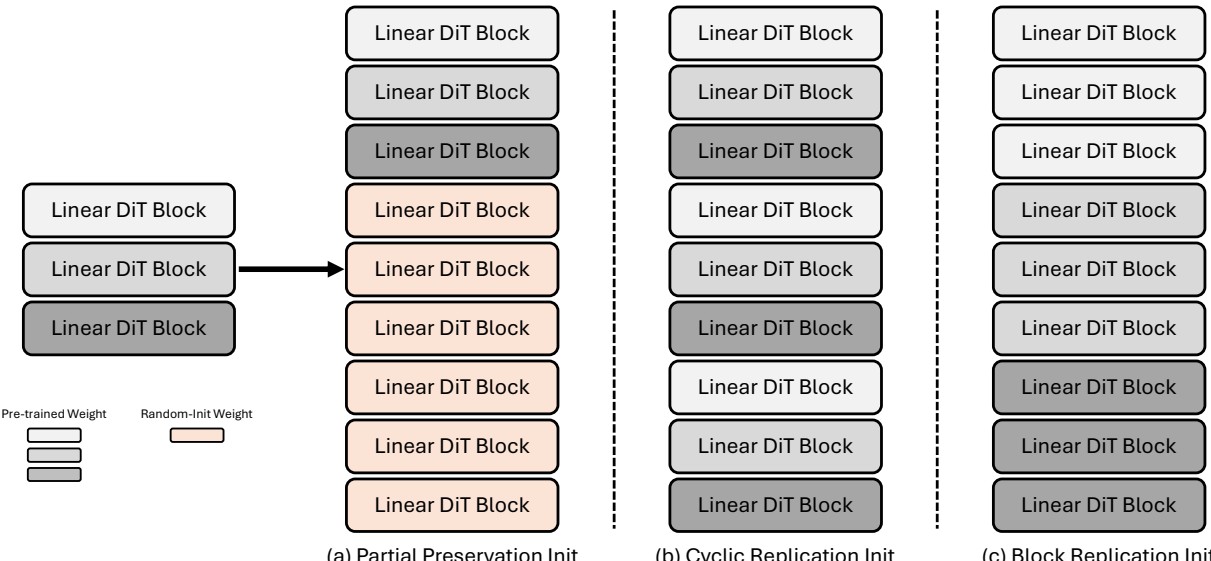

*Figure 11.* **Illustration of initialization strategies.** (a) Partial Preservation Init, which preserves the pre-trained layers and randomly initialize the new layers. (b) Cyclic Replication Init, which repeats the pre-trained layers periodically. (c) Block Replication Init, which extends each pre-trained layer into consecutive layers.

from Mask2Former along with attributes such as color, spatial location, quantity, and various relationships. Importantly, our prompts do not overlap with the GenEval test set. To evaluate the generated images, we leverage the GenEval toolkit. Finally, we train NVILA using the prompts, images, and their corresponding labels (yes or no).

**Training Setup.** We follow the setting in NVILA with learning $2 \times 10^{-5}$, Adam optimizer, cosine scheduler with warmup ratio 0.03 and batch size of 8 per device and train the 2M dataset with one epoch.

# C. More Implementation Details

**Attention with QK Norm** As shown in Figure 10, we introduce RMS normalization (Zhang & Sennrich, 2019) to Query and Key in both linear attention's self-attention block and vanilla cross-attention module to stabilize the training of large diffusion models. Similar to the findings in (Esser et al., 2024), we observe that the attention logits in ReLU-based linear attention (Cai et al., 2023) also grow uncontrollably and frequently exceed the numerical range of FP16 precision (6.5e5), which leads to training instability (NaN). By incorporating QK normalization, we effectively address this issue in large linear transformers. Notably, although our pretrained SANA-1.0 1.6B was not initially trained with QK normalization, we also find that it quickly adapts to these additional normalization layers within just 1K fine-tuning step (Esser et al., 2024). This modification, combined with bf16 mixed precision and our proposed CAME-8bit optimizer (Section 2.3), enables efficient scaling of linear transformer models while maintaining training stability.

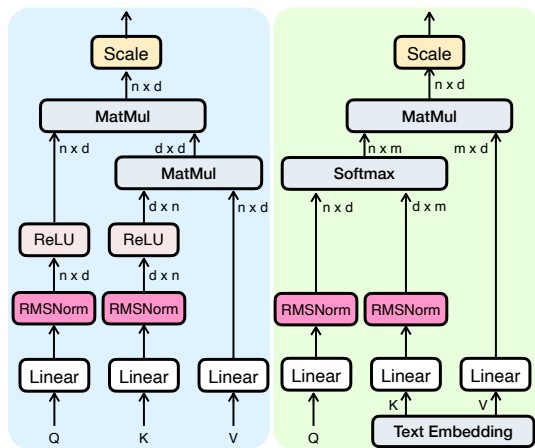

(a). Linear self attention  (b). Vanilla cross attention

*Figure 10.* **Architecture design of linear self-attention and cross-attention blocks in SANA.** Both attention blocks incorporate RMSNorm on query and key for training large model more stable, where linear self-attention is used for content encoding and vanilla cross-attention for text condition injection.

**Multilingual Auto-labeling Pipeline** In Figure 13, we present the results of our multilingual multi-caption auto-labeling pipeline. For each image, we use GPT-4 to translate small-scale data, only 100K English prompts, into: pure Chinese, English-Chinese mixed, and emoji-enriched text. This approach enables us to build a comprehensive multilingual training

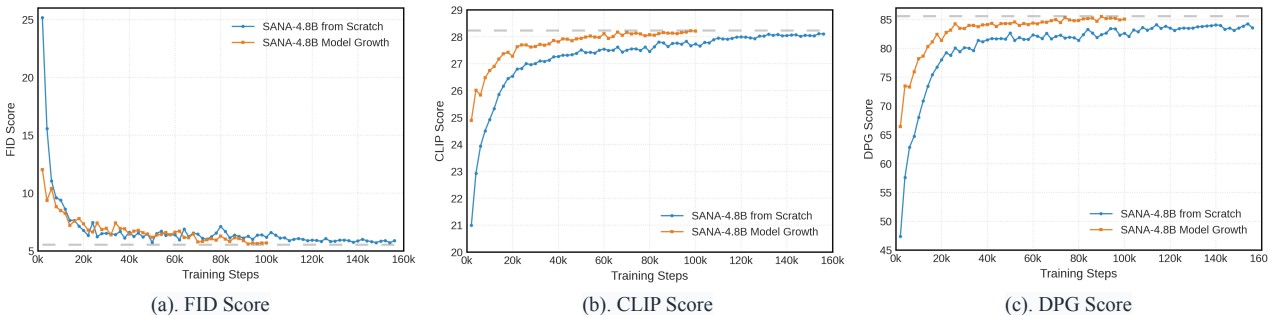

(a). FID Score         (b). CLIP Score         (c). DPG Score

*Figure 12.* **Performance comparison of model growth and training from scratch across FID, CLIP score, and DPGBench metrics.** Our model growth strategy demonstrates superior performance over training from scratch, achieving either better results within the same training duration or equivalent performance with approximately 60% less training time.

dataset that captures diverse ways of describing the same visual content. More results are shown in Figure 14. As a result, we fine-tune SANA with only a few iterations(∼10K), then it demonstrates more stable and accurate outputs for Chinese text and emoji expressions.

**Comparison of Different Initialization Strategies** We illustrate the three types of initialization strategies in Figure 11. Partial Preservation Init preserves the pre-trained layers and randomly initialize the new layers (Figure 11(a)). Cyclic Replication Init repeats the pre-trained layers periodically (Figure 11(b)). Block Replication Init extends each pre-trained layer into consecutive layers (Figure 11(c)). Among these strategies, we adopt the partial preservation initialization approach for its simplicity and stability. Empirically, this approach provides the most stable training dynamics compared to cyclic and block expansion strategies, as shown in Figure 7.

# D. More Results

**Model Growth Results** As shown in Figure 12, our model growth strategy consistently outperforms training from scratch across FID, CLIP score, and DPG benchmarks. Specifically, our approach achieves better quality within the same training duration or reaches equivalent quality with an approximately 60% reduction in training time compared to training from scratch.

**Comparison between different pruned model sizes** As shown in Figure 16, we compare different sizes of SANA-1.5 and SANA-1.0 models. Starting from SANA-1.5 4.8B model (GenEval score 0.693), our pruned variants maintain strong accuracy with 3.2B (0.684) and 1.6B (0.672) parameter counts, consistently outperforming SANA-1.0 1.6B (0.665). This flexible pruning approach allows us to obtain models of any desired size while preserving quality. In particular, larger models demonstrate superior capabilities in image details, pixel quality, and semantic alignment.

**More Visualization Images** In Figure 18, we show more images generated by our model with various prompts. SANA demonstrates comprehensive generation capabilities across multiple aspects, including high-fidelity detail rendering, accurate semantic understanding, and reliable text generation. The samples showcase the model's versatility in handling diverse scenarios, from intricate textures and complex compositions to accurate text rendering and faithful prompt interpretation. These results highlight the robust image quality of the model in both artistic and practical generation tasks.

**More Inference-Time Scaling Examples** We provide additional inference-time scaling examples in Figure 17. During the tournament, VLM judges and filters prompt-mismatching images. We highlight winners with bold green lines and include the winning rationale. Images with incorrect object counts (e.g., the 4th image in Figure 17b) lose the comparison and are filtered by VLM. When two images match the prompt with similar quality, VLM fairly judges that "Both images match the prompt" as shown in Figure 17a and selects one based on preference. Therefore, SANA-1.5 inference scaling effectively filter out those "bad" generations and improves GenEval scores.

**Prompt Rewrite Enhancement** As discussed in (Wang et al., 2024; Han et al., 2024), at inference time, we employ GPT-4o to rewrite user prompts by adding more details, which leads to richer and more detailed visualization results. This

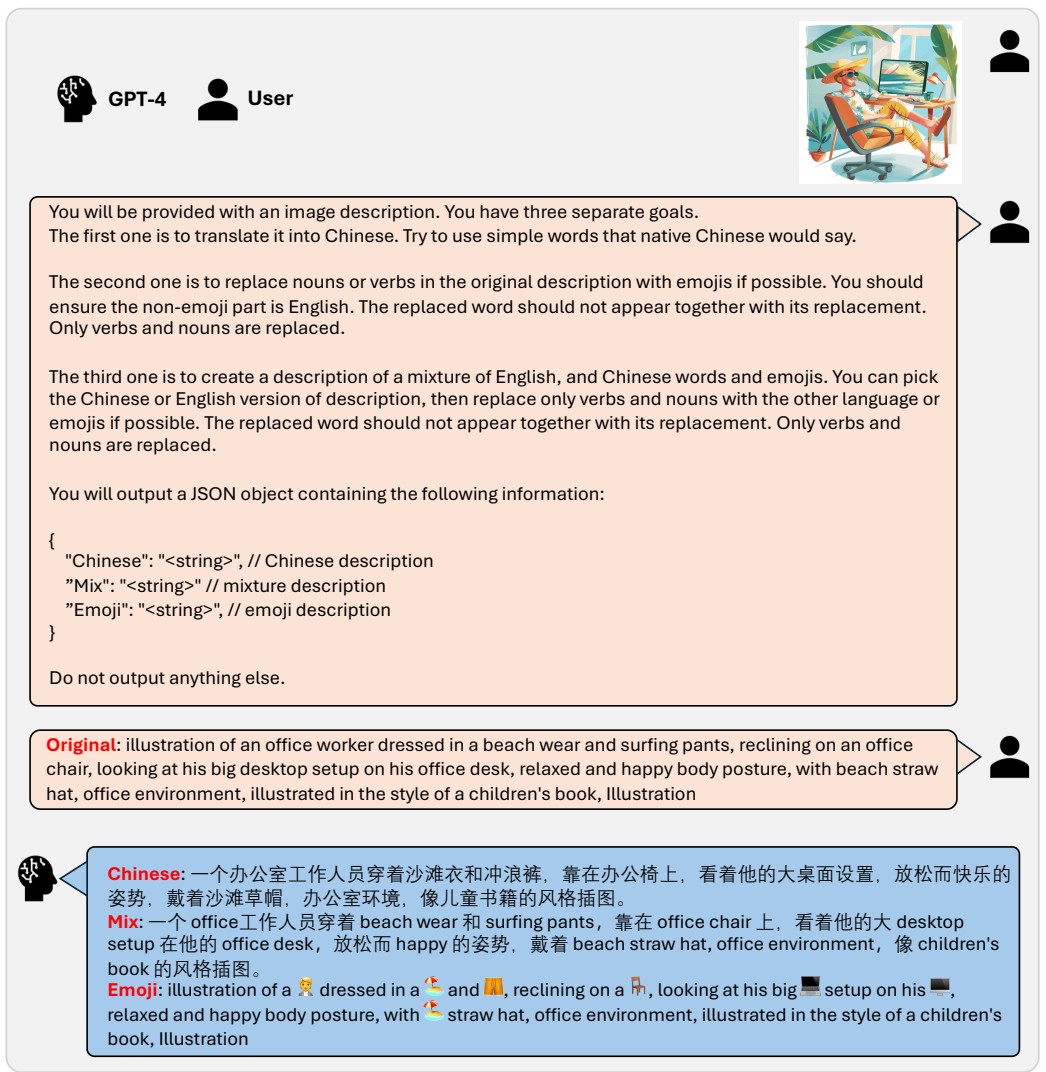

*Figure 13.* **Illustration of our multi-lingual prompt translation pipeline.** We leverage GPT-4 to translate 100k English prompts into four formats: (1) Pure English (2) Pure Chinese (3) English-Chinese mixture (4) Emoji-mixed prompts. Example shows a single English prompt translated into these parallel versions, demonstrating how we construct our multi-lingual training data.

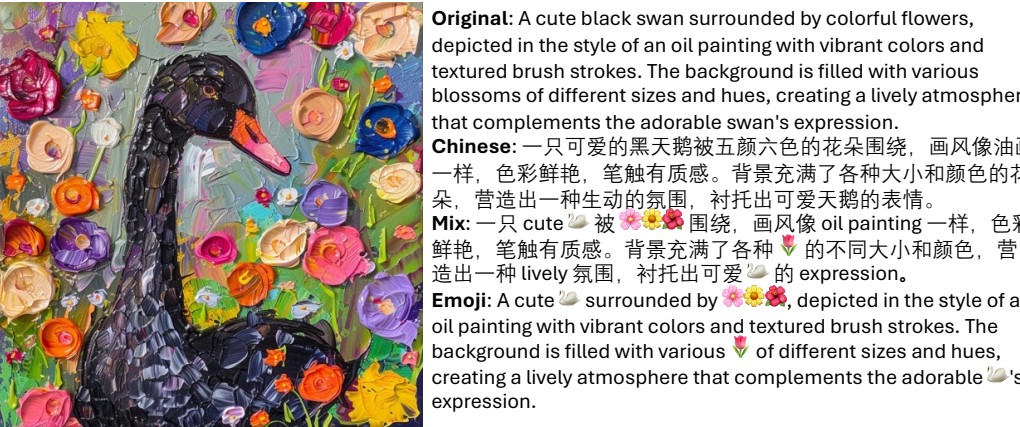

**Original**: A cute black swan surrounded by colorful flowers, depicted in the style of an oil painting with vibrant colors and textured brush strokes. The background is filled with various blossoms of different sizes and hues, creating a lively atmosphere that complements the adorable swan's expression.
**Chinese**: 一只可爱的黑天鹅被五颜六色的花朵围绕，画风像油画一样，色彩鲜艳，笔触有质感。背景充满了各种大小和颜色的花朵，营造出一种生动的氛围，衬托出可爱天鹅的表情。
**Mix**: 一只 cute 🦢 被 🌸🌼🌺 围绕，画风像 oil painting 一样，色彩鲜艳，笔触有质感。背景充满了各种 🌷 的不同大小和颜色，营造出一种 lively 氛围，衬托出可爱 🦢 的 expression。
**Emoji**: A cute 🦢 surrounded by 🌸🌼🌺, depicted in the style of an oil painting with vibrant colors and textured brush strokes. The background is filled with various 🌷 of different sizes and hues, creating a lively atmosphere that complements the adorable 🦢's expression.

*Figure 14.* **More illustration of the multi-lingual dataset.**

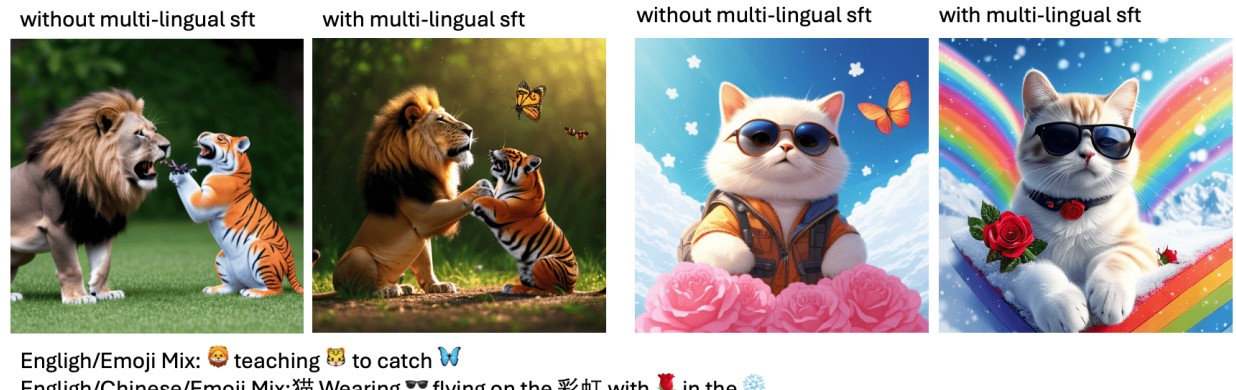

Engligh/Emoji Mix: 😎 teaching 🐯 to catch 🦋
Engligh/Chinese/Emoji Mix:猫 Wearing 🕶 flying on the 彩虹 with 🌹 in the ❄

*Figure 15.* **SANA's multi-lingual capabilities unlocked through efficient fine-tuning.** Comparing image generation quality between baseline model (left, English-only training) and our model (right, fine-tuned with 100k multi-lingual samples) on mixed English/Chinese/emoji prompts.

demonstrates the importance of prompt engineering in maximizing model capabilities. The comparisons are shown in Figure 19.

## E. Discussion of Potential Misuse of SANA-1.5

Misusing generative AI models to generate NSFW content is a challenging issue for the community. To enhance safety, we have equipped SANA-1.5 together with a safety check model (e.g., ShieldGemma-2B (Zeng et al., 2024b)). Specifically, the user prompt will first be sent to the safety check model to determine whether it contains NSFW(not safe for work) content. If the user prompt does not contain NSFW, it will continue to be sent to SANA-1.5 to generate an image. If the user prompt contains NSFW content, the request will be rejected. After extensive testing, we found that ShieldGemma can perfectly filter out NSFW prompts entered by users under strict thresholds, and our pipeline will not create harmful AI-generated content.

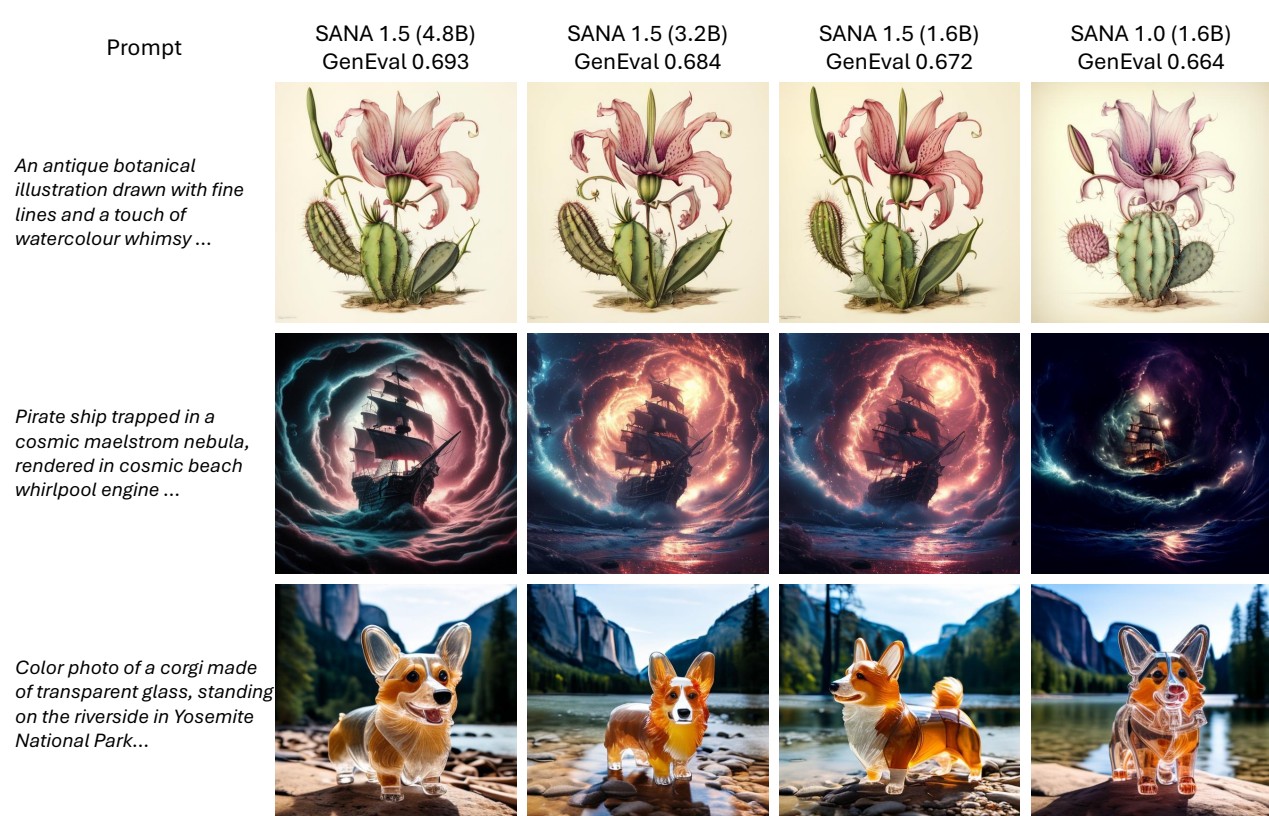

*Figure 16.* **Comparison among different sizes of SANA 1.5 and SANA 1.0.** With model scaling and pruning, SANA 1.5 achieves better performance than SANA 1.0 of the same size, while maintaining flexibility in model capacity selection. Larger models demonstrate enhanced capabilities in detail rendering, image quality, and semantic alignment.

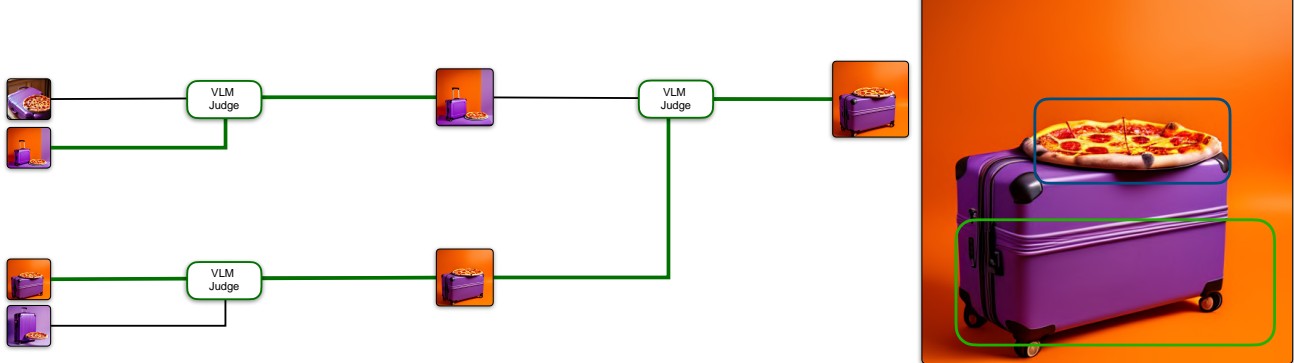

(a) **Prompt**: a photo of a purple suitcase and an orange pizza

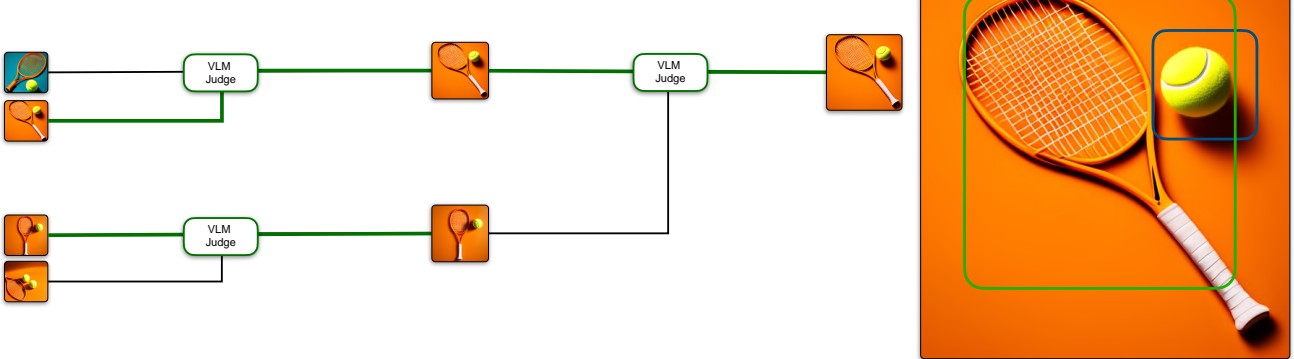

(b) **Prompt**: a photo of an orange tennis racket and a yellow sports ball

*Figure 17.* **Visualization of SANA-1.5 inference-time scaling.** During the tournament, VLM judges and filters prompt-mismatching images.

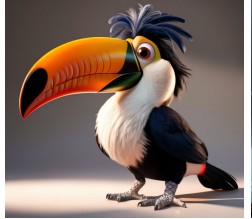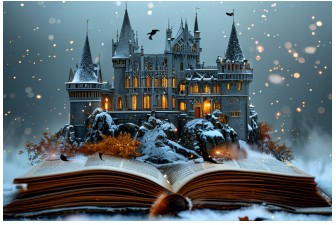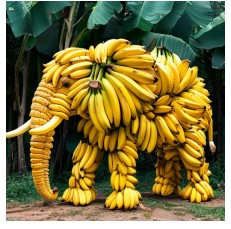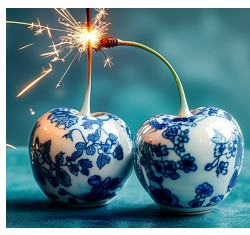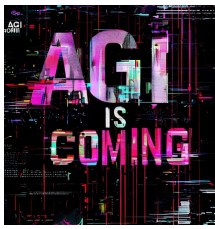

1. A charming 3D-rendered cartoon toucan with exaggerated features, styled in a whimsical yet detailed manner. The character features a disproportionately large, bright orange beak that dominates its face, complemented by a single oversized expressive brown eye. Its plumage combines classic black and white coloring - a fluffy white chest and belly area contrasts beautifully with sleek black feathers covering its back and wings. The bird sports a distinctive messy crest of navy blue feathers on top of its head, giving it a playful, disheveled appearance. The texturing is remarkably detailed, showing individual feathers and subtle variations in the plumage. The character stands on thin, sturdy legs with detailed scaled texture. The lighting setup creates depth and dimension, casting soft shadows that emphasize the bird's round, appealing form against a neutral gradient background. The overall design strikes a perfect balance between cartoon stylization and realistic texturing, making it suitable for animation or game character design

2. A magical castle emerges from the pages of an ancient open book, creating an enchanting winter tableau. The Gothic-style castle, with its soaring spires and numerous towers, stands majestically atop a snow-covered rocky outcrop. The castle's gray stone walls are illuminated by warm golden light pouring from countless arched windows and doorways, creating a welcoming glow against the misty twilight sky. The architecture features intricate details: pointed turrets capped with snow, ornate flying buttresses, and delicate Gothic windows. Small flags flutter from the highest towers, while ravens soar dramatically against the moody sky, their silhouettes adding movement to the scene. In the foreground, a weathered leather-bound book lies open, its pages aged and filled with text in an elegant script. Autumn leaves and snowflakes scatter across its pages, bridging the gap between the real world and the magical realm rising from its pages. Soft bokeh lights dance in the background, suggesting magical sparkles or floating lanterns. The color palette combines cool blues and grays of the winter scene with warm amber glows from the castle windows and scattered lighting elements. The mood is enhanced by a subtle fog that wraps around the castle's base, creating an ethereal atmosphere. The composition suggests a story coming to life, where fantasy and reality merge in a single, spellbinding moment.

3. Big elephant made of bananas

4. still life photography of two porcelain cherries with blue and white floral pattern, Chinese pottery style decoration, one cherry stem has a glowing sparkler, magical sparks against turquoise background, high-end product photography, soft diffused lighting, clean composition, delicate ceramic texture, fine detailed Ming-style floral prints, crisp shadows, glossy ceramic surface reflections, festive celebratory mood, studio lighting setup, sharp focus, luxury product aesthetic

5. poster design with huge text "AGI IS COMING" rendered in glitch art style, digital distortion effects, fragmented typography, corrupted data aesthetics, black ink base with RGB channel splits, cyberpunk graffiti elements, scattered binary code, broken pixel artifacts, CRT scan lines, datamosh effects, quantum computing visual motifs.

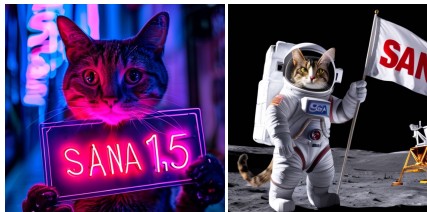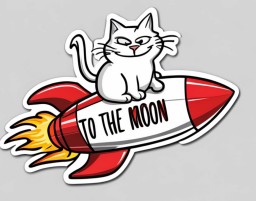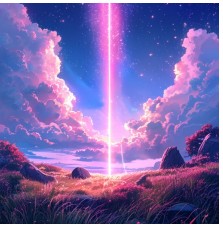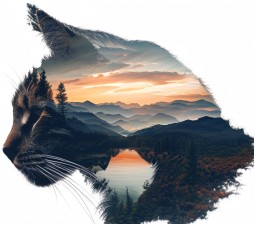

1. a cyberpunk cat with a neon sign that says "SANA 1.5"

2. SANA's meme cosmic adventure continues! Our Tabby cat astronaut, clad in a sleek white spacesuit, stands proudly on the lunar surface. With one paw raised, she holds aloft a flag bearing the iconic SANA logo - The flag flutters in the vacuum of space, its message clear: 'SANA' boldly emblazoned across its fabric. In the background, a small lunar lander rests on the gray, rocky terrain, its ladder extended. A second astronaut, also in a white spacesuit, floats nearby, tethered to the lander. The Earth looms large in the distance, its blue and white surface a stark contrast to the black void of space. This whimsical scene captures the spirit of exploration and the unexpected - a Tabby cat embarking on a mission to the moon, representing the SANA's meme brand in this cosmic setting.

3. A whimsical illustration of the Smirking Cat (the mischievous meme cat) sitting atop a retro-style red and white rocket with text 'TO THE MOON'. The cat appears with its iconic sly grin and playful expression, giving a knowing smirk that perfectly matches the optimistic message. The rocket design is reminiscent of 1950s space age artwork, featuring stylized flame trails in orange and yellow. The entire image has a sticker-like quality with white borders and a minimalist color palette. The composition captures the cat's cheeky, self-satisfied expression, creating a perfect blend of humor and crypto-culture optimism.

4. anime style digital art, mystical light beam piercing through sky, Studio Ghibli inspired landscape, serene meadow with wild grass and rocks, fluffy cumulus clouds, starry twilight sky, pink and purple ethereal light pillar, detailed grass field, soft natural lighting, peaceful countryside scene, vibrant color palette, dreamy atmosphere, fantasy elements, detailed vegetation, cinematic composition

5. surreal art of landscape morphing into a cat's head silhouette, intricate nature elements forming feline features, mountains create pointed ears, rolling hills shape the face curves, forests and meadows texture the fur pattern, rivers flow as whiskers, lakes form the eyes with sunset reflection, clouds define soft fur edges, minimal white background highlighting the cat head shape, dreamlike composition, delicate details, organic flowing lines, nature meets feline anatomy, soft muted color palette, ethereal atmosphere, conceptual illustration style, fine art quality

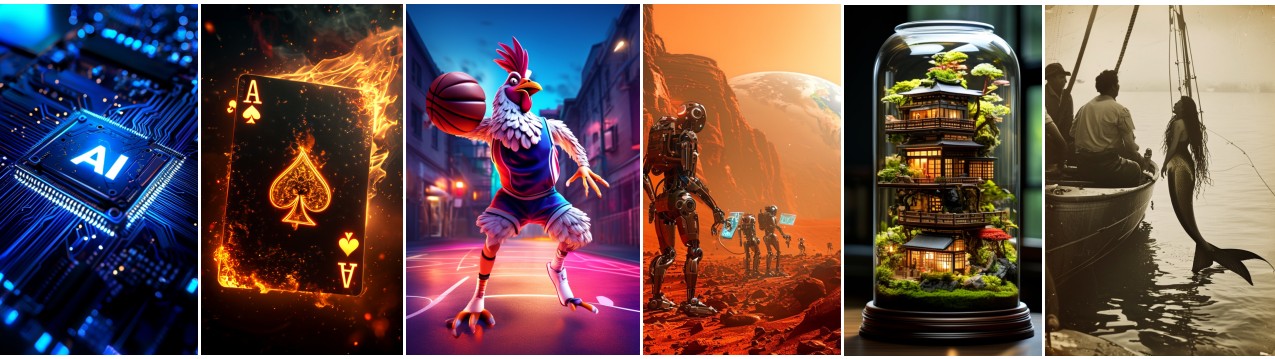

1. ultra-detailed 3D render of premium semiconductor chip, prominent glowing 'AI' text embossed in center, cyberpunk blue illuminated letters against dark metallic surface, futuristic circuit board patterns radiating from text, floating holographic AI letters with light scatter effect, premium macro photography, clean white wave background, depth of field blur, advanced technological aesthetics with precise circuit traces, volumetric lighting around text, ray-traced reflections

2. ace of spades playing card engulfed in dramatic flames, black card with glowing golden symbols, intense fire effects surrounding card edges, burning ember particles, dark background, realistic fire texture, cinematic lighting, high contrast photography, floating sparks and ash, mystical atmosphere, detailed flame tendrils, casino noir aesthetic, high-end 3D render quality, dramatic composition

3. hyperrealist 3D render of a stylized cartoon chicken in basketball uniform, dynamic slam dunk pose, jumping high with basketball, urban street court background, dramatic action shot, motion blur effects, neon lights at dusk, streetball atmosphere, detailed feather texture, glowing court lines, floating movement, cinematic sports photography style, vibrant complementary colors

4. A highly detailed and realistic scene of human-like robots exploring the surface of Mars. The robots have advanced humanoid designs with metallic bodies, glowing sensors, and flexible joints. They are examining the Martian landscape with futuristic equipment, such as scanning devices and holographic displays. The Martian environment includes red rocky terrain, towering cliffs, and a hazy orange sky, with Earth visible faintly in the background. The image captures a sense of discovery and technological advancement, emphasizing the robots' human-like features. Designed in a cinematic, futuristic style

5. ultra-detailed glass bottle terrarium featuring a miniature 3-story traditional Japanese villa, architectural photography style, precise architectural details with wooden beams and tiled roofs, delicate moss and tiny plants growing organically on the structure, ambient interior lighting casting warm glows through miniature shoji screens, the entire scene captured inside a clean cylindrical glass vessel, soft living room lighting enhancing glass reflections, tilt-shift photography effect emphasizing the miniature scale, photorealistic rendering with attention to glass refraction and natural materials

6. Mermaid caught by fishing boat in the early morning, vintage photo from the late 20th century. Men on board are looking at a mermaid-like woman with long hair and a fish tail emerging from the water. She is standing up. Raw, vintage photograph

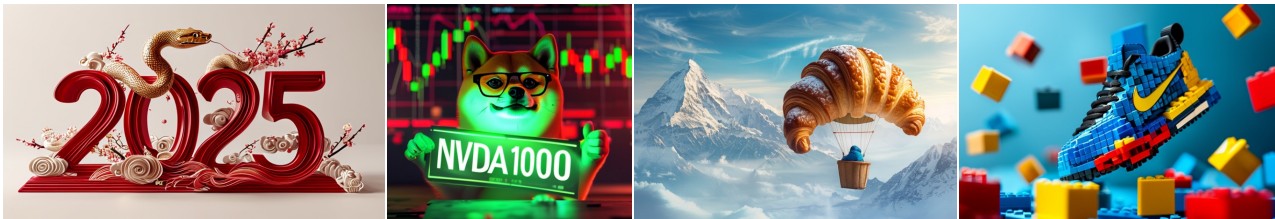

1. A 3D rendering of a red and gold "2025" number with dragon head decoration, adorned with Chinese New Year elements, with a pure white background, studio lighting, soft lighting, and bright colors. This high-resolution photography style adopts minimalist design, with simple lines and details

2. A smirking Shiba Inu (doge meme) wearing nerdy tech glasses, standing on its hind legs and holding a glowing holographic sign that reads "NVDA 1000". The doge has that characteristic knowing grin and raised eyebrows. The scene is set against a background of floating stock charts and red/green candles. The doge wears a simple tech company t-shirt, giving a thumbs up with its free paw. The composition has a meme-like quality with clean, vibrant colors and slight depth of field blur in the background. The holographic sign emits a subtle green glow, matching NVIDIA's brand color.

3. surreal digital art of giant croissant as hot air balloon, person in blue winter jacket riding in woven wicker basket, majestic snow-capped mountain peaks background, soft morning light, crisp pastry details with golden-brown flaky layers, whimsical transportation concept, dreamy winter landscape, ethereal atmosphere, cinematic composition, photorealistic pastry texture, adventure mood, clean blue sky with wispy clouds, high-end food photography meets landscape, magical realism style

4. Magical realism styleproduct photography of Nike sneaker constructed entirely from building blocks, vibrant color blocks in blue, yellow, red and black, floating above scattered toy bricks, intricate brick texture details, modern sneaker silhouette, precise brick construction, macro photography style, shallow depth of field, soft studio lighting, floating brick pieces in background, crisp focus on shoe details, playful toy aesthetic meets product design, dynamic floating composition

*Figure 18.* **High-resolution image generation examples from SANA 1.5**, showcasing its capabilities in the accurate prompt following, spatial reasoning, text rendering, and aesthetics across different styles and aspect ratios.

**Original (Left)**: a photo of two sheep.

**Rewrite (Right)**: A pastoral scene captured in vivid photographic detail. Two woolly ovine graze peacefully in a lush, verdant meadow, their fleecy coats gleaming in the golden afternoon sunlight. The sheep stand side by side, one with a slight tilt of its head as if pausing to regard the camera with mild curiosity. Their large, soulful eyes convey a sense of gentle tranquility, a timeless serenity found in the simple rhythms of nature. The grassy knoll upon which they stand is a patchwork of emerald and sage hues, interspersed with delicate wildflowers in shades of lavender and buttercup yellow. In the distance, the silhouettes of rolling hillsides recede into a hazy azure horizon, creating a bucolic, pastoral tableau. This photographic portrait captures the inherent dignity and peaceful grace of these woolly companions.

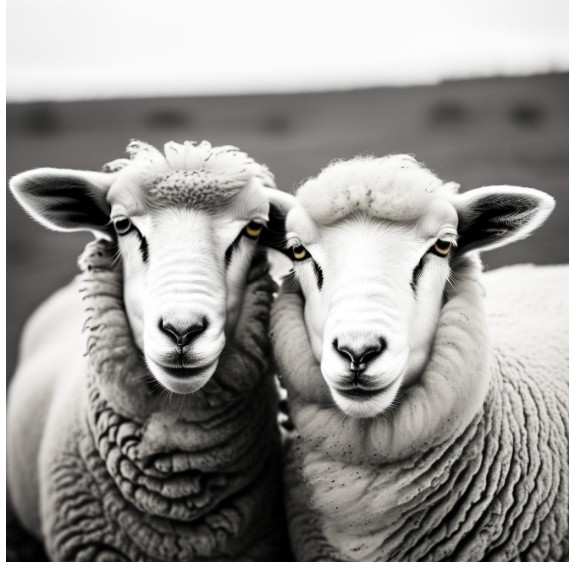 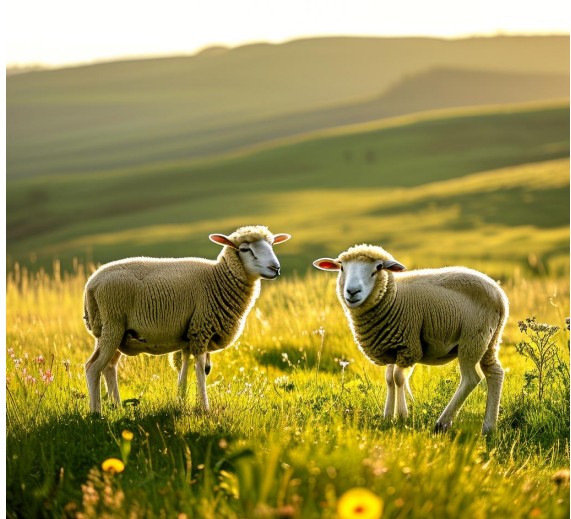

**Original (Left)**: a bench.

**Rewrite (Right)**: A tranquil garden vignette framed by a rustic wooden bench weathered by the elements. The bench's sturdy slats stretch in gently curving lines, their surface worn smooth by the passage of countless visitors seeking respite. Verdant vines and lush flowering plants spill over the edges, softening the bench's rigid form with trailing tendrils and bursts of pastel petals. Dappled sunlight filters through the canopy overhead, casting a warm glow and creating a serene, inviting atmosphere for quiet contemplation. The simple, elegant design of the bench serves as an understated yet essential focal point, beckoning the viewer to pause, sit, and immerse themselves in the calming natural ambiance.

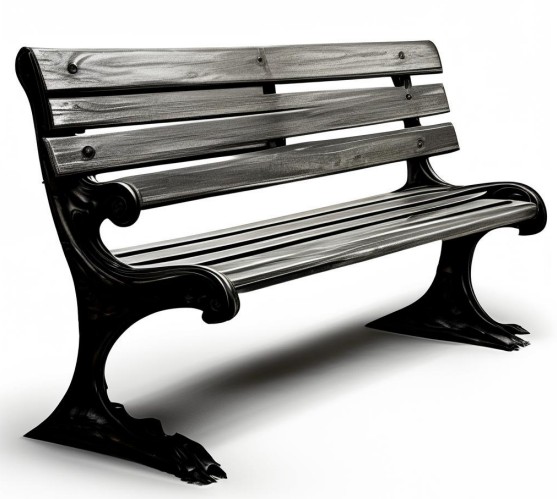 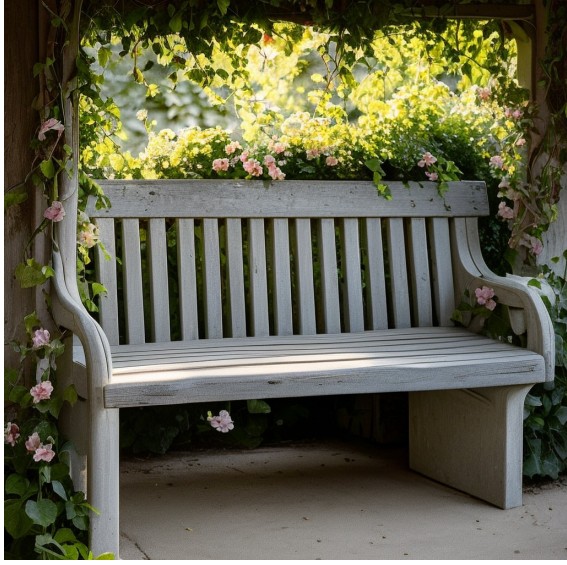

*Figure 19.* **Visual comparison of image generation results before and after prompt enhancement.** For each example, the left shows the result from the original simple prompt, while the right demonstrates the output with enhanced prompt, showing improved visual quality and richer details.

