# OpenReview forum: "SANA 1.5: Efficient Scaling of Training-Time and Inference-Time Compute in Linear Diffusion Transformer"
_ICML.cc/2025/Conference — ICML 2025 poster_

### Official Review · Reviewer_Z15v · 2025-03-12

**Overall Recommendation:** 4

**Summary:**

This paper introduces SANA 1.5, aims at scaling up the linear diffusion transformers from the training and inference perspectives. Based on a pre-trained linear diffusion model. This paper proposes three techniques, Efficient Training Scaling, Model Depth Pruning, and Inference-time Scaling to investigate the scaling question. Each technique is simple and effective. Besides, to further reduce the GPU memory cost during the large model training, SANA 1.5 proposes a memory-efficient optimizer CAME-8bit, and demonstrates its validity. Experimental results show SANA 1.5 is able to surpass some larger models on different metrics.

## update after rebuttal
I appreciate that the authors take time to explain the reason of choosing a deeper rather than a wider model, the experimental results of a from scratch training 20 layers model to address my concerns. And the discussion regarding whether removing the last two layers is instructive. The rebuttal has fully addressed my concerns. I will improve my score and support its acceptance.

**Claims And Evidence:**

The paper claims that both training and inference have scaling properties, but it only show the results of one model scale (1.8B) directly trained from the original 1.6B model. And this paper does not give the log-linear scaling curve between the sampling time (or other metric) v.s. the GenEval (or other metric).

**Essential References Not Discussed:**

No

**Experimental Designs Or Analyses:**

1. For the model configuration, the paper uses the same channel dimension and FFN dimension for different model size (Line 247-Line251), which is not a common choice.
The author claims that directly train the larger models with this configuration from scratch is worse than that using the paper proposed techniques. Thus, is it possible that, using a more common model configuration, the model training from scratch is comparable or even better than the SANA 1.5?
2. In table 3, what is the GenEval result of directly train the model with 3.2B parameter using the technique of Efficient Model Growth?
3. As written in Line315, the author removes the last two blocks. What if do not removing these two blocks and adding new M layers before these layers?

**Methods And Evaluation Criteria:**

Yes

**Other Comments Or Suggestions:**

No

**Other Strengths And Weaknesses:**

Strength:
1. The paper is well written and easy to follow.
2. The problems and contents studied in this paper are of practical application value.
3. The performance is good.
Weakness: Besides the contents in Claims And Evidence and Experimental Designs Or Analyses
1. In figure5(c), it seems that the importance score of newly added layers in SANA 4.8B is significant lower than the original layers in SANA 1.6B. Besides, in figure5(b), the importance scale distribution of SANA-4.8 B is  very different with the SANA 1.6B in figure5(a). A natural question is: Does the model of figure5 b and c trained to convergent?

**Questions For Authors:**

See Other Strengths And Weaknesses

**Relation To Broader Scientific Literature:**

This paper is related to the recently popular test time scaling and accelerated diffusion model.

**Theoretical Claims:**

N/A, no proofs in this paper.

---

> ### Author Rebuttal · Authors · 2025-03-30
>
> ### We sincerely appreciate your insightful feedback, thoughtful comments and kind words. We have carefully addressed all questions to the best of our ability and hope the revisions meet your expectations.
>
> ### Q1: Scaling Depth v.s. Scaling FFN / channel dimensions
> **A1:** While conventional approaches typically scale both depth and width, **SANA-1.5 focuses exclusively on depth scaling to fully utilize existing learned representations.** Recent literature [1] supports this observation, demonstrating that for a fixed parameter budget, deeper models outperform wider architectures [2]. Our experiments in Figure 2 confirm this advantage, showing 60% faster convergence compared to training from scratch.
>
> [1] Wu C, Gan Y, Ge Y, et al. Llama pro: Progressive llama with block expansion[J]. arXiv preprint arXiv:2401.02415, 2024.
>
> [2] Petty J, van Steenkiste S, Dasgupta I, et al. The impact of depth on compositional generalization in transformer language models[J]. arXiv preprint arXiv:2310.19956, 2023.
>
> ---
>
> ### Q2: 3.2B model's GenEval Results if Trained from Scratch
> **A2:** Direct training of 30/40/60-layer models is prohibitively expensive and we are unable to finish training due to limited rebuttal time window (**typically requiring more than two weeks**). However, we have conducted a rigorous comparison between growth-then-prune and train-from-scratch approaches using our 1.6B-parameter (20-layer) models as a representative case study.
>
> The results, shown in the table below, demonstrate that even with identical training configurations, the 1.6B model derived from our 4.8B growth-then-prune pipeline consistently outperforms its directly-trained counterpart across multiple metrics, e.g., FID (**28.67→29.08**), CLIP score (**28.67→29.08**), and GenEval (**0.66→0.67**). Notably, the SANA-1.0 1.6B is fully trained, and further training will not improve the metrics.
>
> We believe these benefits would scale consistently to larger configurations (30/40/60 layers), as the fundamental advantages of our approach - superior initialization and selective compression - remain architecture-agnostic. The computational savings (60% training cost reduction) further demonstrate our method's practical superiority for scalable model training.
>
> | Method        | FID  | CLIP  | GenEval | DPG  |
> |---------------|------|-------|---------|------|
> | SANA-1.0 1.6B | 5.76 | 28.67 | 0.66    | 84.8 |
> | SANA-1.5 1.6B | 5.70 | 29.08 | 0.67    | 84.8 |
>
> ---
>
> ### Q3: Is the Model of Figure.5 (b) and \(c\) Trained to Convergent?
> **A3:** As shown in Figure 2 and Appendix Figure 11, the training curves for GenEval, FID, CLIP-Score, and DPG-Bench all **confirm stable model convergence** throughout training. The block importance score variations originate from our layer-specific initialization strategies (Section 3.2), not from convergence problems.
>
> ---
>
> ### Q4: As Written in Line315, the Author Removes the Last Two Blocks. What if Do not Removing These Two Blocks and Adding New M Layers Before These Layers?
> **A4:** That is a good discussion. We share the insights behind our design: considering that new blocks use **identical mapping initialization**, adding new M blocks after 20 blocks (1.6B) model should be same as adding new M blocks between 18 blocks and the last two blocks. However, when adding new layers without removing last two blocks, they will tend to keep original representation, and hardly to learn from scaling. By removing the last two blocks, the balance of distribution is broken and it will encourage the new layers to learn from scaling.
>
> Our block importance analysis in the table below reveals that initializing from all 20 blocks results in near-zero importance scores. Following our importance metric definition - where higher values indicate more learned knowledge - these results empirically validate our claim (Lines 155-158): **removing and adding will encourage the newly added layers to easily learn new knowledge when scaling model size**.
>
> |Average Block Importance | 1-10 Blocks | 11-20 Blocks | 21-30 Blocks | 31-40 Blocks | 41-50 Blocks | 51-60 Blocks |
> | -------- | -------- | -------- | -------- | -------- | -------- | -------- |
> | Initialize from 20 Blocks |6e-3 | 7e-4 | 2e-7 | 2e-7 | 1e-7 | 1e-7 |
> | Initialize from 18 Blocks | 4e-3 | 5e-4 | 3e-5 | 2.5e-5 | 2e-5 | 2e-5|

---

### Official Review · Reviewer_mxDw · 2025-03-12

**Overall Recommendation:** 3

**Summary:**

This paper proposes three important components: 1. efficient model scaling with depth, 2. efficient model depth pruning, and 3. inference time model performance scaling with VLM. Those three components show that an efficient training strategy can enable a smaller model to achieve performance comparable with the larger models for linear DIT.

**Claims And Evidence:**

Yes

**Essential References Not Discussed:**

Recently, there is one work demonstrated that around 300M on-device model can also achieve 0.55 GenEval score on the device. This might also be a good work to refer.

**Experimental Designs Or Analyses:**

The experimental designs are sufficient.

**Methods And Evaluation Criteria:**

Yes

**Other Comments Or Suggestions:**

NAN

**Other Strengths And Weaknesses:**

Strengths:

This paper proposes how to scale efficient training and inference for linear DiT and make it comparable with the much larger models. The experiments efficiently show that the proposed method is effective.

Weakness:
It might not be a weakness, but this paper proposes those methods without novelty but works pretty well.

**Questions For Authors:**

For the inference-time scaling, how is the speed compared with the larger models with the same quality.

**Relation To Broader Scientific Literature:**

Previous works have demonstrated that linear transformer can be applied well on the T2I generation, but this paper summarizes previous works on scaling efficient linear DiT.

**Theoretical Claims:**

Yes, they all look good to me.

---

> ### Author Rebuttal · Authors · 2025-03-30
>
> ### We sincerely appreciate your insightful feedback, thoughtful comments and kind words. We have carefully addressed all questions to the best of our ability and hope the revisions meet your expectations.
>
> ### Q1: Efficiency-Cost Tradeoff of VLM verifier
> **A1:** This is a good question and we list the FLOPs of each model
>
> * SANA 0.6B model: N x 261G*20(steps)=N x 5,220G
> * SANA 1.6B model: N x 704G*20=N x 14,040G
> * SANA 4.8B model: N x 2457G*20=N x 49,140G
> * NVILA-verfier-2B: (N-1) x 4,518G
>
> where $N$ is the numer of generated images. We notice that SANA-0.6B with 32 generation scaling is better than SANA-4.8B without verification in both performance and FLOPS.
>
> | Method+Samples          | FLOPs    | GenEval |
> |-------------------------|----------|---------|
> | SANA-0.6B + 16 images   | 151,290G | 0.75    |
> | SANA-4.8B + 4 images    | 196,560G | 0.72    |
>
> ---
>
> ### Q2: Add more good references
> **A2:** Thanks for noticing the reference we should have. We will add SnapGen[1] (please correct if you mean other projects) to our efficiency related work.
>
> [1] Hu D, Chen J, Huang X, et al. SnapGen: Taming High-Resolution Text-to-Image Models for Mobile Devices with Efficient Architectures and Training[J]. arXiv preprint arXiv:2412.09619, 2024.

---

### Official Review · Reviewer_fWyY · 2025-03-13

**Overall Recommendation:** 4

**Summary:**

The paper introduces SANA-1.5, which incorporates a series of techniques to efficiently scale up the SANA-1.0 linear-attention diffusion Transformer. Specifically, it proposes a depth-growth paradigm that includes partial-layer preservation initialization and a memory-efficient CAME-8bit optimizer. Additionally, the paper presents a grow-then-prune-and-tune approach to reduce the number of layers and an inference-time scaling strategy that leverages VLM models. The proposed paradigm demonstrates a highly efficient scaling path, significantly reducing training compute while maintaining image generation quality.

**Update after rebuttal**: Based on the reviews, rebuttal, and discussions, I am keeping my final rating as an Accept. In addition to the strengths highlighted in my initial review, the additional information and experiments provided in the rebuttal further solidify the contributions of this paper.

**Claims And Evidence:**

The paper’s central claim regarding the high training and inference costs of existing visual generation models is both critical and practical, particularly as large-scale models become essential for high-quality generation yet remain inaccessible to many practitioners due to computational constraints. This work is well-motivated, and the proposed paradigm offers a comprehensive and feasible solution well-supported by empirical results. It addresses key research questions, such as "How is the scalability of linear diffusion Transformer?" and "How can we scale up large linear DiT and reduce the training cost?", and presents a practical and inspiring approach with implications for future research and applications.

**Essential References Not Discussed:**

N/A.

**Experimental Designs Or Analyses:**

Please see my comment in Methods And Evaluation Criteria above. Overall, I am very satisfied with the thorough experiments on each argument and design choice made in this paper.

**Methods And Evaluation Criteria:**

The proposed techniques, partial-layer preservation initialization, the memory-efficient CAME-8bit optimizer, the grow-then-prune-and-tune approach, and inference-time scaling, are all well-motivated and highly effective. Together, they form a comprehensive framework for efficiently scaling up the model.

The evaluation primarily consists of detailed analyses and ablations on various design choices, including different initialization strategies, comparisons between the CAME-8bit optimizer and CAME, AdamW, and AdamW-8bit, as well as the impact of high-quality data fine-tuning, model block importance, and the grow-prune-tune approach on computational cost and output quality. This thorough and systematic analysis makes the findings compelling. Additionally, the paper includes comparisons with existing methods on widely applied benchmarks such as GenEval, DPG, and MJHQ, clearly demonstrating the effectiveness of the proposed techniques.

**Other Comments Or Suggestions:**

N/A.

**Other Strengths And Weaknesses:**

N/A.

**Questions For Authors:**

Only two non-critical questions:
1. How is the grow-then-prune approach compared to directly training a 40/30/20 blocks model with the same initialization methods?
2. Quantitative experiments on stability enhancement and identity mapping initialization could further support the claims in those sections.

**Relation To Broader Scientific Literature:**

This work builds upon existing literature and effectively integrates a diverse range of techniques into a comprehensive system. It extends prior works such as SANA-1.0 (Xie et al., 2024), DiT (Peebles & Xie, 2022), CAME (Luo et al., 2023), and block-wise 8-bit quantization (Dettmers et al., 2021), as well as various techniques detailed in Appendix A, including diffusion model pruning and training and inference scaling. The paper precisely cites and references relevant works. Overall, this work presents a novel and well-grounded approach.

**Theoretical Claims:**

Apart from the proposed techniques discussed earlier, which are well-supported by experiments, other key claims and design choices, such as the block-wise quantization strategy and the hybrid precision design of the CAME-8bit optimizer, as well as the distinction between inference scaling and training scaling, are highly reasonable. I did not identify any critical correctness issues in the claims presented in this paper, at least within the reasonable scope of this paper.

---

> ### Author Rebuttal · Authors · 2025-03-30
>
> ### We sincerely appreciate your insightful feedback, thoughtful comments and kind words. We have carefully addressed all questions to the best of our ability and hope the revisions meet your expectations.
>
> ### Q1: Growth-then-Prune vs Train-from-Scratch
> **A1:** Direct training of 30/40/60-layer models is prohibitively expensive and we are unable to finish training due to limited rebuttal time window (**typically requiring more than two weeks**). However, we have conducted a rigorous comparison between growth-then-prune and train-from-scratch approaches using our 1.6B-parameter (20-layer) models as a representative case study.
>
> The results, shown in the table below, demonstrate that even with identical training configurations, the 1.6B model derived from our 4.8B growth-then-prune pipeline consistently outperforms its directly-trained counterpart across multiple metrics, e.g., FID (**28.67→29.08**), CLIP score (**28.67→29.08**), and GenEval (**0.66→0.67**). Notably, the SANA-1.0 1.6B is fully trained, and further training will not improve the metrics.
>
> We believe these benefits would scale consistently to larger configurations (30/40/60 layers), as the fundamental advantages of our approach - superior initialization and selective compression - remain architecture-agnostic. The computational savings (60% training cost reduction) further demonstrate our method's practical superiority for scalable model training.
>
>
> | Method        | FID  | CLIP  | GenEval | DPG  |
> |---------------|------|-------|---------|------|
> | SANA-1.0 1.6B | 5.76 | 28.67 | 0.66    | 84.8 |
> | SANA-1.5 1.6B | 5.70 | 29.08 | 0.67    | 84.8 |
>
> ---
>
> ### Q2: Stability Quantification for Different Initalizating Strategies
> **A2:** As demonstrated in Figure 7 and the accompanying table, the initialization strategy critically impacts training stability. Our experiments show that suboptimal initialization frequently causes training instability, manifesting as: (1) numerical instability (NaN values) during optimization, and (2) failure to produce meaningful quantitative results.
>
>
> | Method                    | 3k Iterations Training Loss |
> |---------------------------|-----------------------------|
> | Cyclic Replication Init   | NaN                         |
> | Block Replication Init    | NaN                         |
> | Partial Preservation Init | 0.835                       |
>
> The identity mapping initialization we employ represents the current standard practice for model growth scenarios, as evidenced by[1-3]. This initialization scheme effectively addresses the vanishing/exploding gradient problems that commonly plague direct training of large architectures, while maintaining the model's representational capacity at the beginning of the growth process.
>
>
> [1]Han S, Mao H, Dally W J. Deep compression: Compressing deep neural networks with pruning, trained quantization and huffman coding[J]. arXiv preprint arXiv:1510.00149, 2015.
>
> [2]Chen, T., Goodfellow, I., and Shlens, J. Net2net: Accelerating learning via knowledge transfer. arXiv preprint arXiv:1511.05641, 2015.
>
> [3]Karras T, Aila T, Laine S, et al. Progressive growing of gans for improved quality, stability, and variation[J]. arXiv preprint arXiv:1710.10196, 2017.

---

> > ### Comment · Reviewer_fWyY · 2025-04-05
> >
> > Thanks to the authors for responding to my questions and providing additional results that support the claims more. I don't have any further questions, and I will decide the final rating based on all the reviews, rebuttals, and discussions. Thanks.

---

### Official Review · Reviewer_nKKe · 2025-03-14

**Overall Recommendation:** 4

**Summary:**

The paper introduces SANA-1.5, an efficient linear Diffusion Transformer for text-to-image generation.

Key contributions include:

(1) A depth-growth paradigm that scales models from 1.6B to 4.8B parameters, reducing training costs by 60%;

(2) A technique called model depth pruning via block importance analysis for flexible compression;

(3) An inference-time scaling strategy using VLM selection.

The experimental results show the proposed model's superior performance.

## update after rebuttal
Thank the authors for their rebuttal. I have no more concerns and would like to keep my score (4: Accept).

**Claims And Evidence:**

Most claims made in the paper are generally supported by clear and convincing observations and experiments.

The authors have presented sufficient qualitative and quantitative results to support their main contributions.

One concern is that more results and experiments are required. to verify the effectiveness of the proposed inference-time scaling strategy.

**Essential References Not Discussed:**

No.

**Experimental Designs Or Analyses:**

The experimental designs are generally sound and align well with its claims.
Evaluations on benchmarks such as GenEval, FID, and CLIP score support the results, with practical metrics like memory usage and efficiency included.
It would be better if the authors can provide some qualitative comparison with other models based on the same text prompts.
Moreover, the the VLM-based inference-scaling method need to be further verified.

**Methods And Evaluation Criteria:**

The proposed methods are well-designed and appropriate for address text-to-image generation.

It provided some insight to push the boundaries of this fields.

**Other Comments Or Suggestions:**

- It would be better if you can align the styles of different tables and figures for a more cohesive and polished presentation.
- Using "Inference Scaling Law" in Line 370 is a little bit overclaimed.
- Incorporating human evaluations based on some real-world noisy or long text prompts would be better.

**Other Strengths And Weaknesses:**

Strengths:

- The overall improvement is intuitive, which also maintains the elegance of the framework.
- The writing is easy-to-following. The figures and tables are all good.
- It is good to see that the overall image quality of pruned models can be easily recovered after brief fine-tuning (Figure 4).

Weaknesses:
- Please see other sections for weaknesses.

**Questions For Authors:**

- Could you provide more quantitative results to show the claims of Figure 3? Why the adopted VLM-based method is best and in what scenarios(or let's say, situations) is it the best scheme?
- Generating multiple images per prompt increases inference costs. Have you quantified the additional computational overhead compared to the efficiency gains achieved by using a smaller model? Is there a clear analysis of this trade-off?
- Have you considered extending the whole paradigm to other fields like text-to-video or text-to-3D generation? Are there any further different aspects that need to be further dealed with?

**Relation To Broader Scientific Literature:**

No.

**Theoretical Claims:**

This paper doesn't contain any complex proofs and the math notations are correct so far.

---

> ### Author Rebuttal · Authors · 2025-03-30
>
> ### We sincerely appreciate your insightful feedback, thoughtful comments and kind words. We have carefully addressed all questions to the best of our ability and hope the revisions meet your expectations.
>
> ### Q1: Will VLM-based Verifier be the Best and any Other Choices?
>
> **A1:** We choose VLM-based verifier mainly because the advantages in **context length and flexible number of images**. In the L252-256 of our submission, we discussed that
> > While popular models like CLIP (Radford et al., 2021) and SigLIP (Zhai et al., 2023) offer multi-modal capabilities, their small context windows (77 tokens for CLIP and 66 for SigLIP) limit their effectiveness. This limitation poses a problem since SANA requires long, detailed descriptions.
>
> We want the verifier to be more general and can be scaled up to long descriptions and multi-images thus choose VLM as the backbone. The performance shows ideal scaling curve in our Figure 8.
>
> ---
>
> ### Q2: More Quantitative of VLM-based Inference Time Scaling
>
> **A2**: While we have presented the quantitative results in Table 2 and Figure 8, we further **applied the inference-time scaling to SD1.5 to validate** the generalization capability.
>
> We have iterate a version of VLM-verifier with more data and better training setting and with the new VLM-verifier, we achieved a remarkable improvement in **SD1.5's GenEval score from 0.42 to 0.87**. When combined with our more advanced SANA-1.5 4.8B model (0.72 GenEval), this approach reaches **0.95** GenEval score, setting a new state-of-the-art on GenEval.
>
> | Method                  | **Overall** | Single | Two  | Counting | Colors | Position | Color Attribution |
> |-------------------------|---------|--------|------|----------|--------|----------|-------------------|
> | SD1.5              | **0.42**    | 0.98   | 0.39 | 0.31     | 0.72   | 0.04     | 0.06              |
> | SD1.5 + Inference Scaling     | **0.87**    | 1.00   | 0.97 | 0.93     | 0.96   | 0.75     | 0.62              |
> | SANA-1.5 4.8B        | **0.72**    | 0.99   | 0.85 | 0.77     | 0.87   | 0.34     | 0.54              |
> | SANA-1.5 4.8B + Inference-Time Scaling     |   **0.95**        | 1.00 | 1.00 | 0.95 | 0.96 | 0.91 | 0.85
>
> ---
>
> ### Q3: Efficiency-Cost Tradeoff of VLM verifier
> **A3:** This is a good question and we list the FLOPs of each model
>
> * SANA 0.6B model: N x 261G*20(steps)=N x 5,220G
> * SANA 1.6B model: N x 704G*20=N x 14,040G
> * SANA 4.8B model: N x 2457G*20=N x 49,140G
> * NVILA-verfier-2B: (N-1) x 4,518G
>
> where $N$ is the numer of generated images. We notice that SANA-0.6B with 16 generation scaling is better than SANA-4.8B without verification in both performance and FLOPS.
>
> | Method+Samples          | FLOPs    | GenEval |
> |-------------------------|----------|---------|
> | SANA-0.6B + 16 images   | 151,290G | 0.75    |
> | SANA-4.8B + 4 images    | 196,560G | 0.72    |
>
> ---
>
> ### Q4: Writing Suggestion and Visualization
> **A4:** Thanks to the reviewer's carefullness, we will revise accordingly:
>
> * We will change the title to from "Inference Scaling Law" to "Inference Time Scaling".
> * We will polish the tables and figures using consistent style for better presentation.
> * We will add a visualization figure comparing with other SoTA methods with the same prompts, including some real-world noisy or long text prompts. We prepare a preview for all reveiwers here: https://www.filemail.com/d/xgaxjwwlpdjttkx
>
> ---
>
> ### Q5: Extension to Video/3D
> **A5:** This is an exciting direction we're exploring. Specifically, we can directly add an additional temporal attention layer in each block and further fine-tune with video data, while preserving the linear attention block and cross attention blocks. In this way, we can better utilize our pre-trained text-to-image model and try to make the training of text-to-video model more efficient. We'll try our best to give more answers in our future study in these fields.

---

> > ### Comment · Reviewer_nKKe · 2025-04-05
> >
> > Good to see your sufficient and professional rebuttal response that has addressed my concerns.
> >
> > I still have some questions for further discussion. The proposed test-time scaling method seems to be very intuitive yet effective. Can you provide more in-depth explanations for this, regarding:
> > -  Why could such a method actually work well? Have you considered any more complex paradigm to assess generated images?
> > - What are the advantages and limitations of such a paradigm compared to scaling up the model parameters?
> > - Does the VLM play a critical role here? Were alternative VLMs tested?

---

> > > ### Author Response · Authors · 2025-04-07
> > >
> > > ### Q6: In-depth Explanations for Proposed Intuitive yet Effective Test-Time Scaling Method
> > >
> > > **A6**:
> > > ### 1. In-depth Explanations
> > > During our evaluation of generative models, we observed that when processing compositional prompts (e.g., “a red cube on top of a blue sphere”), the model’s output quality depends on the initial random noise. Certain noise samples lead to semantically correct generations, while others result in failures (e.g., object misplacement or missing attributes). This aligns with prior studies on noise sensitivity in diffusion models:
> > >
> > > **Golden Noise[1]**: Demonstrates that diffusion models are sensitive to initial noise, and optimizing noise selection can improve output fidelity.
> > >
> > > **Noise Optimization for Diffusion Models[2]**: Proposes training auxiliary networks to predict "high-quality" noise.
> > >
> > > **On the Role of Initial Noise in Diffusion Sampling[3]**: Analyzes how noise statistics affect compositional reasoning.
> > >
> > > Stronger models (e.g., larger or better-trained variants) are more robust to noise variability, generating correct outputs more frequently across different noise samples.
> > >
> > > ### 2. Our Approach: Test-Time Scaling via Output Selection
> > >
> > > Instead of pre-selecting "golden noise" at the input stage (which requires additional training or heuristics), we propose generating multiple outputs with diverse noise samples and using a verifier (e.g., CLIP or VLM feedback) to select the best output. This approach is more scalable and avoids the computational overhead of noise optimization, while leveraging the observation that high-capacity models intrinsically produce more correct samples under noise variation.
> > >
> > > ### 3. Technical Advantages:
> > >
> > > **Noise-agnostic**: Eliminates dependency on noise-selection models.
> > >
> > > **Computationally efficient**: Shifts complexity to inference-time validation.
> > >
> > > **Model-agnostic**: Applicable to any diffusion-based architecture.
> > >
> > > ### 4. Future Plans
> > >
> > > Our observations and preliminary experiments suggest that more sophisticated paradigms will be beneficial for advancing inference-time scaling. Specifically, we envision leveraging multi-modal validation frameworks to generalize and enhance output selection strategies. Below we outline potential future directions:
> > >
> > > 1. Hybrid Evaluation Paradigms: Future work could integrate detection models (e.g., YOLO, DETR) with rule-based verification (GenEval-like rules) to dynamically assess output quality.
> > > 2. Combining VLMs (e.g., GPT-4V, LLaVA) and domain-specific models could provide hierarchical validation. For example, use VLMs for high-level semantic alignment (e.g., "Does the image match the prompt?"), then apply precise detector-based methods to checks for critical attributes.
> > >
> > > [1] Zhou Z, Shao S, Bai L, et al. Golden noise for diffusion models: A learning framework[J].
> > >
> > > [2] Qi Z, Bai L, Xiong H, et al. Not all noises are created equally: Diffusion noise selection and optimization[J].
> > >
> > > [3] Ban Y, Wang R, Zhou T, et al. The Crystal Ball Hypothesis in diffusion models: Anticipating object positions from initial noise[J].
> > >
> > > ---
> > >
> > > ### Q7: Advantages and Limitations compared to Model Scaling
> > >
> > > **A7**: Compared to model size scaling, our approach offers distinct advantages: Training larger models incurs higher costs and greater training difficulties. In contrast, our verifier-based inference-time scaling method is more generalizable and orthogonal to model scaling. It improves different models (e.g., SD and SANA) and can be combined with larger models for better performance.
> > >
> > > Notably, the effectiveness of inference-time scaling further improves performance when combined with larger and more capable base models. Since the verifier operates as a relatively independent component, it introduces no fundamental limitations - users can train custom verifiers tailored to their specific requirements.
> > >
> > > To our knowledge, the primary limitation lies in computational efficiency: Larger models inherently may have slower inference speeds, and incorporating inference-time scaling further increases computational costs due to the need for multiple generations and verifications.
> > >
> > > ---
> > >
> > > ### Q8: Does the VLM play a critical role here? Were alternative VLMs tested??
> > > **A8**: Yes, VLM is critical for inference-time scaling to boost performance. As shown in the table in our previous response Q2, scaling with VLM effectively boosts **SD1.5's GenEval score from 0.42 to 0.87** and **SANA-1.5 4.8B model from 0.72 to 0.95**. Additionally, in response to mxDW's Q1, we demonstrate that **SANA-1.6B + VLM Scaling can generate better images with less FLOPs than SANA-4.8B**.
> > >
> > > We haven not tested other VLMs because existing open-source and commercial VLMs cannot accurately assess image-prompt matching (discussed in Section 2.5), thus  finetuning is necessary. **We choose NVILA because it provides the finetuning docs and abilities**, while other VLMs like Qwen-VL and InternVL do not officially offer the feature.

---

### Decision · Program_Chairs · 2025-05-01

**Decision:**

Accept (poster)

**Comment:**

SANA-1.5 introduces an efficient linear DiT for text-to-image generation, incorporating techniques such as efficient training scaling, depth pruning, and inference scaling. The paper presents promising generation results with significantly reduced training costs. The authors actively engaged in rebuttal and discussion with reviewers. All reviewers unanimously voted for acceptance. The paper is recommended for acceptance, and the authors are encouraged to include the reviewer-author discussion in the final version.